# Model Uncertainty Quantification by Conformal Prediction in Continual Learning

**Rui Gao** [1] **Weiwei Liu** [1]

## Abstract

Continual learning has attracted increasing research attention in recent years due to its promising experimental results in real-world applications. In this paper, we study the issue of calibration in continual learning which reliably quantifies the uncertainty of model predictions. Conformal prediction (CP) provides a general framework for model calibration, which outputs prediction intervals or sets with a theoretical high coverage guarantee as long as the samples are exchangeable. However, the tasks in continual learning are learned in sequence, which violates the principle that data should be exchangeable. Meanwhile, the model learns the current task with limited or no access to data from previous tasks, which is not conducive to constructing the calibration set. To address these issues, we propose a **CP**-based method for model uncertainty quantification in **c**ontinual **l**earning (CPCL), which also reveals the connection between prediction interval length and forgetting. We analyze the oracle prediction interval in continual learning and theoretically prove the asymptotic coverage guarantee of CPCL. Finally, extensive experiments on simulated and real data empirically verify the validity of our proposed method.

## 1. Introduction

As vast amounts of data are produced, the number of new tasks is increasing overwhelmingly (Zou & Liu, 2023b; Liu et al., 2019; Gong et al., 2023b; Chen & Liu, 2023). Consequently, learning systems are required to rapidly adapt to the continuously emerging tasks (Gong et al., 2023a; 2021; 2022; Xu & Liu, 2022; Zou & Liu, 2023a). Continual learning (CL)—also referred to as lifelong learning (Chen & Liu, 2018; Gao & Liu, 2025b), sequential learning (Aljundi et al., 2019), and incremental learning (Aljundi et al., 2018)—is a paradigm that enables a model to learn a large number of tasks sequentially, where data from previous tasks are no longer accessible during the training of new tasks. A significant challenge in continual learning is catastrophic forgetting (Kumaran et al., 2016), where standard deep learning methods tend to rapidly forget previously acquired knowledge when learning new tasks (Kirkpatrick et al., 2017). Most works of continual learning have concentrated on mitigating this issue. For example, the regularization-based methods such as EWC (Kirkpatrick et al., 2017) and MAS (Aljundi et al., 2018) measure the parameter importance and introduce a regularization term in the loss function to consolidate previous knowledge; replay-based methods such as iCaRL (Rebuffi et al., 2017) and GEM (Lopez-Paz & Ranzato, 2017) try to replay the knowledge of previous tasks by storing samples or generating pseudo-samples, and then provide these replayed knowledge to the model while a new task is being learned; parameter isolation methods such as PackNet (Mallya & Lazebnik, 2018) and HAT (Serrà et al., 2018) dedicate different model parameters to each task so as to prevent any possible forgetting. Overall, these works mitigate catastrophic forgetting of model in continual learning, which is often reflected as improved accuracy of model on previous tasks.

However, existing works to date have ignored the issue of calibration in continual learning. Calibration is often deemed as important as the standard criterion of accuracy in statistics and machine learning (Thelen et al., 2022; Park et al., 2024). A well-calibrated model can reliably quantify the uncertainty of its prediction (Guo et al., 2017; Hermans et al., 2021). Model uncertainty quantification is crucial in many applications of continual learning. For example, in medical imaging, advancements in technology or changes in diagnostic procedures result in continuous variations in image appearance (Hofmanninger et al., 2020; Ghesu et al., 2021). In this context, the model must provide critical information to doctors, enabling them to understand the uncertainty in model predictions and make informed decisions regarding patient care.

[1]School of Computer Science, National Engineering Research Center for Multimedia Software, Institute of Artificial Intelligence and Hubei Key Laboratory of Multimedia and Network Communication Engineering, Wuhan University, Wuhan, China. Correspondence to: Weiwei Liu <liuweiwei863@gmail.com>.

*Proceedings of the 42$^{nd}$ International Conference on Machine Learning*, Vancouver, Canada. PMLR 267, 2025. Copyright 2025 by the author(s).

Among various uncertainty quantification techniques, we specifically concentrate on *conformal prediction* (CP) (Vovk et al., 2005; Ndiaye, 2022; Fisch et al., 2022; Qian et al., 2024) in this paper. CP provides a general framework for model calibration, which is model-agnostic, nonparametric and distribution-free (the coverage guarantee holds for any distribution). CP takes in three components: a black-box predictive model, an input feature and a potential output. It requires the design of a score function to measure how non-conforming the potential output is and to calculate scores on a calibration set. In regression tasks, CP outputs prediction intervals while in classification tasks, it produces prediction sets. These prediction intervals or sets can theoretically guarantee high coverage probability as long as the samples are exchangeable (Shafer & Vovk, 2008)—a weaker requirement than the standard i.i.d. assumption. However, uncertainty quantification using CP poses significant challenges in continual learning. As illustrated in Figure 1, tasks are learned by a single model in sequence. Some works have shown that changing the order of tasks significantly affect the performance of model in continual learning (Lange et al., 2022). This violates the principle of data exchangeability. Meanwhile, samples from previous tasks are limited or inaccessible, resulting in a restricted calibration set for CP. Therefore, effective use of CP for model uncertainty quantification in continual learning requires further investigation.

In this paper, we consider the regression setting and construct prediction intervals for continual learning with a asymptotic coverage guarantee. Our main contributions can be summarized as follows:

- We propose a **CP**-based method for model uncertainty quantification in **c**ontinual **l**earning, termed **CPCL**. CPCL constructs a new dataset which takes the dependencies among original data into count and defines the prediction interval by a conditional quantile estimator. Additionally, CPCL reveals the inherent connection between prediction interval length and forgetting.

- We analyze the oracle prediction interval in continual learning and provide the asymptotic coverage guarantee of the prediction interval of CPCL.

- Extensive experimental results on simulated and real data demonstrate the validity of CPCL.

## 2. Related Work

**Continual learning.** Continual learning consists of a sequence of disjoint tasks that are sequentially learned one at a time. After sequentially learning all tasks, continual learning wants the model to perform well on all seen tasks. Existing research on continual learning focuses primarily on

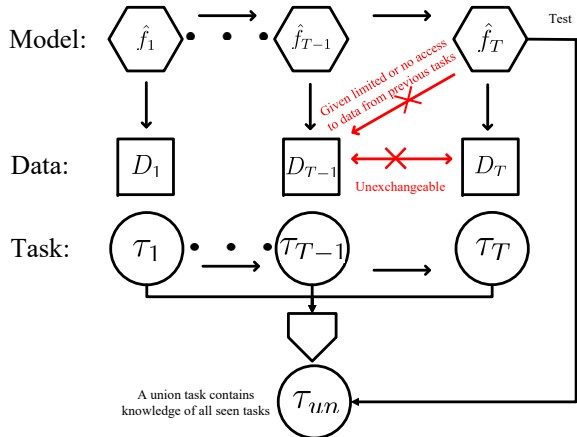

*Figure 1.* A general framework of continual learning.

how to solve the catastrophic forgetting of accuracy. Some surveys (Parisi et al., 2019; Lesort et al., 2019; Pfülb & Gepperth, 2019; Farquhar & Gal, 2018) have shown that existing continual learning works can be broadly divided into three categories: (i) replay methods, (ii) regularization-based methods and (ii) parameter isolation methods.

Replay methods generally have two ways to replay in artificial neural networks: partial replay (PR) and generative replay (GR) (Hayes et al., 2021; Gao & Liu, 2023). Replay methods store samples or generate pseudo-samples using a generative model, and these samples are replayed while a new task is being learned (Rebuffi et al., 2017; Lopez-Paz & Ranzato, 2017; Rolnick et al., 2019; Ayub & Wagner, 2021). PR stores either all or a subset of previously learned inputs in a replay buffer and mixes these inputs with new samples to train the classifier (Rebuffi et al., 2017; Lopez-Paz & Ranzato, 2017; Rolnick et al., 2019; Ayub & Wagner, 2021) while GR generates synthetic samples for previous tasks. For example, iCaRL (Rebuffi et al., 2017) selects and stores samples (exemplars) closest to the feature mean of each class by assuming fixed allocated memory; DER (Buzzega et al., 2020) mixes rehearsal with knowledge distillation and regularization and matches the network's logits sampled throughout the optimization trajectory to promot consistency with its past; DGR (Shin et al., 2017) uses GAN to to generate previous samples for data replaying. In addition to previous task samples, graph-based replay (Tang & Matteson, 2021) have been proven to be efficient.

Regularization-based methods add an extra regularization term to the loss function and penalize changes to important parameters of model for previous tasks (Kirkpatrick et al., 2017; Zenke et al., 2017; Aljundi et al., 2018; Lee et al., 2017; Gao & Liu, 2025a). For example, EWC (Kirkpatrick et al., 2017) uses the Fisher Information Matrix to measure the importance of parameters; IMM (Lee et al.,

2017) estimates Gaussian posteriors for task parameters, in the same vein as EWC, but inherently differs in its use of model merging; MAS (Aljundi et al., 2018) measures the importance according to the gradients of the squared $L_2$-norm of learned network output function. Other kinds of regularization-based methods are also efficient, such as functional regularization (Pan et al., 2020; Benjamin et al., 2019), node importance (Jung et al., 2020) and uncertainty regularization (Ahn et al., 2019).

Parameter isolation methods dedicate different model parameters to each task to prevent any possible forgetting (Xu & Zhu, 2018; Mallya & Lazebnik, 2018; Serrà et al., 2018). PackNet (Mallya & Lazebnik, 2018) iteratively assigns parameter subsets to consecutive tasks by constituting binary masks. HAT (Serrà et al., 2018) incorporates task-specific embeddings for attention masking where the per-layer embeddings are gated through a Sigmoid to attain unit-based attention masks in the forward pass.

**Conformal prediction.** As introduced in Section 1, CP is a model-agnostic, nonparametric and distribution-free framework for the calibration of models. It designs a score function which measures how non-conforming the potential output is and evaluates these scores on a hold-out calibration set. CP produces prediction intervals (or sets) where data is assumed to be exchangeable. Under the assumption of exchangeable data, CP has be successful in many applications. For example, Wisniewski et al. (2020) focus on the application of conformal prediction interval estimations to provide financial Market Makers (MMs) with some "meaningful" forecasts relating to their future short-term position in a given financial market; (Huang et al., 2023) propose conformalized GNN, extending conformal prediction to graph-based models for guaranteed uncertainty estimates; Gui et al. (2023) adapt the framework of conformal prediction to propose a distribution-free method for predictive inference in the matrix completion problem. Some works consider the situations in which the data exchangeability is not satisfied (Tibshirani et al., 2019; Gendler et al., 2022; Xu & Xie, 2023a). Tibshirani et al. (2019) show that a weighted version of conformal prediction can be used to compute distribution-free prediction intervals for problems in which the test and training covariate distributions differ. Gendler et al. (2022) consider that the test data may be adversarially attacked where the exchangeability assumption is grossly violated. Xu & Xie (2023a) develop the general framework for constructing prediction intervals for time series. Compared to time series, continual learning has the following differences when studying CP. First, in time series, samples from previous time points are available, while in continual learning, only samples from the current task are available, and samples from previous tasks are not accessible. This means that a complete calibration set can be naturally constructed in time series, while the calibration set in continual learning is limited to the current task. Second, time series focuses on the model's performance on data arriving at future time points, whereas the core concern of continual learning is the catastrophic forgetting regarding previous tasks. The connection between the prediction intervals (sets) constructed by CP and the catastrophic forgetting in continual learning requires further research. Overall, it is necessary to study CP in continual learning.

## 3. Preliminaries

We mainly follow the continual learning setting of (Delange et al., 2021) and conformal prediction setting of (Xu & Xie, 2023b). In this paper, we consider a regression setup. Given an unknown model, $f : \mathbb{R}^d \to \mathbb{R}$, each observation $Z = (X, Y)$ is generated according to the following model

$$Y = f(X) + \mu \tag{1}$$

where $\mu$ is distributed following a continuous cumulative distribution function (CDF) $F$, $Y$ is a continuous scalar variable, $X \in \mathbb{R}^d$ denotes the feature and $d$ is the dimension of the feature vector.

**Continual learning.** There are $N_\tau$ disjoint tasks $(\tau_1, \tau_2, ..., \tau_{N_\tau})$ that are learned sequentially. The dataset of task $\tau_t$ is given by $\{Z_n^t = (X_n^t, Y_n^t)\}_{n=1}^N$, where $N$ is the number of samples for each task. Each observation $Z_n^t = (X_n^t, Y_n^t)$ for task $\tau_t$ is generated according to Eq.(1) where $\mu_n^t$ is distributed according to $F^t$. In continual learning, the model is trained on the current task while access to data from previous tasks is limited. The goal of continual learning is to control the statistical risk of the model across all previously seen tasks:

$$\sum_{t=1}^{N_\tau} \mathbb{E}_{(X^t, Y^t)}[\ell(h(X^t; \theta), Y^t)] \tag{2}$$

where $\ell$ is the loss function and $h$ is the predictor with parameter $\theta$.

**Conformal prediction.** In the context of CP, there are $N_{cal}$ observable samples $\{Z_i = (X_i, Y_i)\}_{i=1}^{N_{cal}}$ which are generated according to Eq.(1). The goal of CP is to construct a sequence of prediction intervals that are as narrow as possible while ensuring a certain coverage guarantee. Given a user-specified prediction algorithm, we use the observed samples to obtain a trained model represented by $\hat{f}$. Subsequently, we construct a prediction interval $\widehat{C}_{test}^\alpha$ for $Y_{test}$ of a test sample, where $\alpha$ is the significance level. CP considers two types of coverage guarantees. The conditional coverage guarantee ensures that the prediction interval $\widehat{C}_{test}^\alpha$ satisfies:

$$P(Y_{test} \in \widehat{C}_{test}^\alpha | X_{test}) \geq 1 - \alpha. \tag{3}$$

The second type is the marginal coverage guarantee:

$$P(Y_{test} \in \widehat{C}_{test}^{\alpha}) \geq 1 - \alpha. \qquad (4)$$

It is noteworthy that Eq.(3) is significantly stronger than Eq.(4). As established by split conformal prediction (SCP) (Papadopoulos et al., 2007), the marginal coverage guarantee holds whenever data are exchangeable (Definition 1).

**Definition 1.** *(Exchangeability (Shafer & Vovk, 2008)) The random variables $V_1, V_2, \ldots, V_n$ are exchangeable if for any permutation $\gamma$ of integers $1, 2, \ldots, n$, the variables $\overline{V}_1, \overline{V}_2, \ldots, \overline{V}_n$, where $\overline{V}_i = V_{\gamma(i)}$, have the same joint probability distribution as $V_1, V_2, \ldots, V_n$.*

In this paper, we aim to construct prediction intervals with a conditional coverage guarantee through CP in continual learning setting. We consider $N_{\tau}$ disjoint tasks $(\tau_1, \tau_2, ..., \tau_{N_{\tau}})$. The dataset for task $\tau_t$ is $\{Z_n^t = (X_n^t, Y_n^t)\}_{n=1}^{N}$. As the goal of continual learning is to control the statistical risk across all seen tasks (i.e. Eq.(2)), the test sample should derive from a union task that encompasses all knowledge from previous tasks. Our work replaces the test sample $Z_{test} = (X_{test}, Y_{test})$ with $Z_{ut} = (X_{ut}, Y_{ut})$ for convenience, where $Z_{ut} = (X_{ut}, Y_{ut})$ represents the test sample from a union task that contains knowledge from all seen tasks. After sequentially learning all tasks using any continual learning algorithm (which can be viewed as the user-specified prediction algorithm), we obtain a trained model. Given $X_{ut}$, we quantify the uncertainty of the trained model by constructing a prediction interval $\widehat{C}_{ut}^{\alpha}$ for the output, ensuring that $Y_{ut} \in \widehat{C}_{ut}^{\alpha}$ with probability $1 - \alpha$. In light of this, some natural challenges arise:

- The samples from previous tasks are limited when learning the current task $\tau_T$. This limitation implies that the calibration set reserved by typical conformal prediction methods (such as SCP) is also restricted.

- Some works have shown that changing the order of tasks significantly affect the performance of model in continual learning (Lange et al., 2022). Therefore, the principle that data should be exchangeable is violated in continual learning. In this case, the courage of typical conformal prediction methods (such as SCP) cannot be guaranteed (Zou & Liu, 2024).

To address these challenges, we demonstrate how to quantify model uncertainty using conformal prediction for continual learning in Section 4. Additionally, we establish asymptotic conditional validity of our proposed method in Section 5.

## 4. Uncertainty Quantification in Continual Learning

Inspired by Xu & Xie (2023a), we show how to quantify model uncertainty by CP for continual learning in this sec-

tion. We focus on the current task $\tau_T$. After sequentially learning the first $T$ tasks, we obtain a trained model $\hat{f}_T$. For a given input $X_{ut}^T$, we mainly construct a prediction interval $\widehat{C}_{ut}^{\alpha}$ to effectively qualify the potential prediction.

**Calibration set.** As discussed in Section 3, the calibration set is limited at the current task $\tau_T$ (i.e., we only have access to data of current task to construct the calibration set). Inspired by the replay-based methods in continual learning (Rebuffi et al., 2017; Lopez-Paz & Ranzato, 2017; Rolnick et al., 2019; Ayub & Wagner, 2021), we try to replay the samples of previous tasks to construct the calibration set. For simplicity, we maintain a buffer to store $N_{cal}$ samples from each previous task. While other replay techniques could be considered, we do not develop them in this paper. Overall, by the replay, we collect the calibration set defined as $\bigcup_{t=1}^{T} \{Z_i^t = (X_i^t, Y_i^t)\}_{i=1}^{N_{cal}}$.

**Nonconformity score function.** For each observation $Z_i^t = (X_i^t, Y_i^t)$ in the calibration set, we define the prediction error as

$$\hat{\mu}_i^t = Y_i^t - \hat{f}_T(X_i^t). \qquad (5)$$

Then we define the nonconformity score function as

$$s(X_i^t, Y_i^t) = \frac{1}{1 + e^{-\hat{\mu}_i^t}} \qquad (6)$$

which ensures that the scores range in $(0, 1)$. We introduce this nonconformity score function for two reasons. First, the range of sigmoid-based nonconformity score function is $(0, 1)$. Our method is involved with the training of Quantile Regression Forests which requires to construct trees. This process asks us to define that the score function ranges in $(0, 1)$ for rigorous proof. Second, this nonconformity score function is invertible, which helps us to rewrite the prediction interval. Details of reasons can be found in Appendix B. We leverage Eq.(6) to calculate the non-conformity scores for all observations in calibration set $\bigcup_{t=1}^{T} \{Z_i^t = (X_i^t, Y_i^t)\}_{i=1}^{N_{cal}}$. This results in the score set $\bigcup_{t=1}^{T} \{S_i^t\}_{i=1}^{N_{cal}}$, where $S_i^t = s(X_i^t, Y_i^t)$. It is noteworthy that the nonconformity score function defined in this paper is independent of the calibration set. Since observations in continual learning are not exchangeable, the calibration set exhibits inherent dependencies among its observations. Consequently, the scores in $\bigcup_{t=1}^{T} \{S_i^t\}_{i=1}^{N_{cal}}$ should be sequentially dependent because of the independence of the nonconformity score function and the inherent dependencies among calibration observations. To account for these dependencies in the scores, we train a quantile estimator $\hat{\mathcal{Q}}^T(x; \alpha)$ on $\bigcup_{t=1}^{T} \{S_i^t\}_{i=1}^{N_{cal}}$ rather than using the empirical quantile directly.

**Quantile regression forests (QRF) (Meinshausen, 2006).** In this paper, we utilize QRF to achieve the quantile esti-

mator $\hat{\mathcal{Q}}^T(x;\alpha)$. Considering the dependencies over scores, we aim to reconstruct a new dataset based on the score set to train QRF. In this new dataset, each input should encapsulate the knowledge of dependencies over scores and each prediction should represent the conditional quantile of the scores from the union task. Specifically, we define the reconstructed dataset as $D_T^R = \{(X_i^R, Y_i^R)\}_{i=1}^{N_{cal}-1}$, where $X_i^R = [S_i^1, S_i^2, \ldots, S_i^T]$ and $Y_i^R$ is an approximation of the true score $S_{ut}^T$ of the observation $(X_{ut}^T, Y_{ut}^T)$ from the union task which contains knowledge from all seen tasks. It is noteworthy that $X_i^R$ contains $T$ sequential scores which are effective to predict the condition quantile of true score $S_{ut}^T$.

We train QRF on the reconstructed dataset $\{(X_i^R, Y_i^R)\}_{i=1}^{N_{cal}-1}$. After training QRF, we grow $K$ trees. For each tree with separate parameter $\zeta_k$, there are $L$ leaves, where every leaf $l$ is associated with a rectangular subspace $R_l \subset \mathbb{B}$. These subspaces are disjoint and cover the entire space $\mathbb{B}$, i,.e. for any $x \in \mathbb{B}$, there is one and only one leaf which is denoted as $R_{l(x,\zeta_k)}$.

Based on the trained QRF, we then leverage $X_{N_{cal}}^R = [S_{N_{cal}}^1, S_{N_{cal}}^2, \ldots, S_{N_{cal}}^T]$ to predict the conditional quantile of $S_{ut}^T$. Given the reconstructed dataset $\{(X_i^R, Y_i^R)\}_{i=1}^{N_{cal}-1}$, the trained QRF and $X_{N_{cal}}^R \in \mathbb{B}$, we can obtain the estimated conditional distribution function $\hat{F}(y_{N_{cal}}^R | X_{N_{cal}}^R)$ as follows:

$$p_i(X_{N_{cal}}^R, \zeta_k) = \frac{\mathbb{I}(X_i^R \in R_{l(X_{N_{cal}}^R, \zeta_k)})}{|\{j \in [N_{cal}-1] | X_j^R \in R_{l(X_{N_{cal}}^R, \zeta_k)}\}|} \quad (7)$$

$$p_i(X_{N_{cal}}^R) = \sum_{k=1}^K p_i(X_{N_{cal}}^R, \zeta_k)/K \quad (8)$$

$$\hat{F}(y_{N_{cal}}^R | X_{N_{cal}}^R) = \sum_{i=1}^{N_{cal}-1} p_i(X_{N_{cal}}^R) \mathbb{I}(Y_i^R \leq y_{N_{cal}}^R) \quad (9)$$

where $[N_{cal}-1]$ represents the set $\{1, 2, \ldots, N_{cal}-1\}$. $R_{l(X_{N_{cal}}^R, \zeta_k)}$ is the leaf of tree $\zeta_k$ that contains the input $X_{N_{cal}}^R$. If this leaf also contains the observation $X_i^R$ from the reconstructed dataset, then $\mathbb{I}(X_i^R \in R_{l(X_{N_{cal}}^R, \zeta_k)})$ is equal to 1; otherwise, it is equal to 0. $|\{j \in [N_{cal}-1] | X_j^R \in R_{l(X_{N_{cal}}^R, \zeta_k)}\}|$ represents the number of observations included in this leaf from the reconstructed dataset. $p_i(X_{N_{cal}}^R, \zeta_k)$ measures the weight of observation $X_i^R$, while $p_i(X_{N_{cal}}^R)$ further measures the weight over $K$ trees. In practice, we implement the above fitting and prediction process by Python according to (Meinshausen, 2006).

**Prediction interval.** Based on the estimated conditional distribution function $\hat{F}(y_{N_{cal}}^R | X_{N_{cal}}^R)$, we define the conditional quantile estimator $\hat{\mathcal{Q}}^T(x = X_{N_{cal}}^R; \alpha)$ as

$$\hat{\mathcal{Q}}^T(x = X_{N_{cal}}^R; \alpha) = \inf\{s \in (0,1) : \hat{F}(s | X_{N_{cal}}^R) \geq \alpha\}. \quad (10)$$

Then the prediction interval with significance level $\alpha$ is

$$\hat{C}_{ut}^\alpha = \{Y_{ut}^T : \hat{\mathcal{Q}}^T(X_{N_{cal}}^R; \hat{\beta}) \leq s \leq \hat{\mathcal{Q}}^T(X_{N_{cal}}^R; 1-\alpha+\hat{\beta})\} \quad (11)$$

where $s = s(X_{ut}^T, Y_{ut}^T)$. For convenience, we define $\hat{\mathcal{Q}}^T(X_{N_{cal}}^R; \alpha) = \hat{s}_\alpha$. Recalling the nonconformity score function in Eq.(6), we can rewrite the prediction interval as

$$\hat{C}_{ut}^\alpha = [\hat{f}_T(X_i^t) - \ln(\frac{1}{\hat{s}_{\hat{\beta}}}-1), \hat{f}_T(X_i^t) - \ln(\frac{1}{\hat{s}_{1-\alpha+\hat{\beta}}}-1)]. \quad (12)$$

$$\hat{\beta} = \operatorname*{argmin}_{\beta \in [0,\alpha]}(\ln(\frac{1}{\hat{s}_{\hat{\beta}}}-1) - \ln(\frac{1}{\hat{s}_{1-\alpha+\hat{\beta}}}-1)) \quad (13)$$

**Connection between intervals and forgetting.** As the number of tasks increases, the phenomenon of forgetting becomes more pronounced. The model's test error on previous tasks rises (Eq.(5)), while the error on the current task remains low. This leads to an increased difference between our estimated $\hat{s}_{\hat{\beta}}$ and $\hat{s}_{1-\alpha+\hat{\beta}}$. According to Eq.(12), we can conclude that the length of the prediction intervals will also increase, which is verified by Figure 3 in Section 6.2.

**Conformal prediction for continual learning.** Overall, we propose a **CP**-based method for model uncertainty quantification in **c**ontinual **l**earning, termed **CPCL**. Specifically, when learning current task $\tau_T$, we first obtain a trained model $\hat{f}_T$ using a specific continual learning method. Then we construct the calibration set by replaying samples of previous tasks. We collect score set by calculating nonconformity scores on calibration set. Following that, we reconstruct a new dataset based on the score set. Since the union task contains knowledge from all seen tasks, we select one entry $S_i^j$ of $X_i^R$ which is multiplied by a coefficient $\omega$, and set $Y_i^R = S_i^j$. We then train QRF on this reconstructed dataset to obtain conditional quantile estimator. Finally, for any test input $X_{ut}^T$, we produce the prediction interval which comprises the conditional quantile estimator. Details of CPCL are shown in Algorithm 1. In practice, we consider $N_{test}$ test samples and obtain intervals based on these test samples.

## 5. Theory Analysis

In this section, we first analyze the oracle prediction interval in continual learning. Inspired by Meinshausen (2006); Xu & Xie (2023a), we then demonstrate that the conditional distribution function estimated by our proposed CPCL (i.e., Eq.(9)) converges to true conditional distribution function. Finally, we prove that this convergence of distribution function results in the convergence of quantile estimates, indicating asymptotic conditional validity of CPCL. For convenience, we let $T = N_\tau$ represent the scenario where all $N_\tau$ tasks in continual learning have been learned.

**Algorithm 1 CP**-based method for model uncertainty quantification in **c**ontinual **l**earning (CPCL).

---

**Input:** $(\tau_1, \tau_2, ..., \tau_{N_\tau})$: all training tasks; $N_\tau$: total number of tasks; $\{Z_n^t = (X_n^t, Y_n^t)\}_{n=1}^N$: the dataset of task $\tau_t$; $N$: the number of samples for each tasks; $T$: the index of current task; $N_{cal}$: the number of samples from each task in the calibration set; $CLM$: a certain continual learning method.

1: Initialization: $RB = \emptyset$
2: **for** $T \in [1, \ldots, N_\tau]$ **do**
3:    Obtain $\hat{f}_T$ from a certain continual learning method $CLM$.
4:    Collect $\{Z_n^T = (X_n^T, Y_n^T)\}_{n=1}^{N_{cal}}$ from task $\tau_T$.
5:    $RB = RB \cup \{Z_n^T = (X_n^T, Y_n^T)\}_{n=1}^{N_{cal}}$.
6:    Construct $\bigcup_{t=1}^T \{S_i^t\}_{i=1}^{N_{cal}}$ on $RB$ by Eq.(6).
7:    Reconstruct $D_T^R = \{(X_i^R, Y_i^R)\}_{i=1}^{N_{cal}-1}$.
8:    Fit QRF on $D_T^R$ and obtain the conditional quantile estimator by Eq.(10).
9:    Construct the prediction interval $\widehat{C}_{ut,T}^\alpha$ for any test input $X_{ut}^T$ with significance level $\alpha$ by Eqs.(11), (12) and (13).
10: **end for**
**Output:** $(\widehat{C}_{ut,1}^\alpha, \widehat{C}_{ut,2}^\alpha, \ldots, \widehat{C}_{ut,N_\tau}^\alpha)$

---

## 5.1. Oracle Prediction Interval in Continual Learning

Here we discuss the oracle prediction interval in continual learning, which includes $Y_{ut}^{N_\tau}$ with an exact conditional coverage of $1 - \alpha$. We first define $Z^t = (X^t, Y^t)$ as the data originating from task $\tau_t$. The oracle prediction assumes perfect knowledge of $f$ in Eq.(1). Given $X_{ut}^{N_\tau}$, and since $N_\tau$ tasks are learned in sequence during continual learning, the prediction of $Y_{ut}^{N_\tau}$ is conditioned on $X_{ut}^{N_\tau}$ and the sequence $\boldsymbol{Z_1^{N_\tau}} = [Z^1, Z^2, \ldots, Z^{N_\tau}]$. Therefore, the goal of CP in continual learning centers around the CDF of $Y_{ut}^{N_\tau}$ conditioning on $X_{ut}^{N_\tau}$ and the sequence $\boldsymbol{Z_1^{N_\tau}}$, which is defined as:

$$
\begin{aligned}
F_{ut}(y) &= \mathbb{P}(Y_{ut}^{N_\tau} \le y | X_{ut}^{N_\tau}, \boldsymbol{Z_1^{N_\tau}}) \\
&= \mathbb{P}(\mu_{ut}^{N_\tau} \le y - f(X_{ut}^{N_\tau}) | \boldsymbol{Z_1^{N_\tau}})
\end{aligned} \quad (14)
$$

where $Y_{ut}^{N_\tau} = f(X_{ut}^{N_\tau}) + \mu_{ut}^{N_\tau}$ and $\mu_{ut}^{N_\tau}$ is distributed following $F_{ut}^{N_\tau}$. According to Eqs.(5) and (6), our designed score function $s(\cdot)$ is dependent of the data sequence $\boldsymbol{Z_1^{N_\tau}}$. $\boldsymbol{S_1^{N_\tau}}$ is defined as the score sequence in which $S^t = s(Z^t)$. In this case, the dependence among $\boldsymbol{S_1^{N_\tau}}$ arises from $\boldsymbol{Z_1^{N_\tau}}$. Therefore, the conditional CDF in Eq.(14) is equivalent to the following CDF based on the score sequence

$$
F_{ut}(s|\boldsymbol{S_1^{N_\tau}}) = \mathbb{P}(S_{ut}^{N_\tau} \le s | \boldsymbol{S_1^{N_\tau}}). \quad (15)
$$

We define the true conditional quantile as

$$
\mathcal{Q}^{N_\tau}(\alpha) = \inf\{s : F_{ut}(s|\boldsymbol{S_1^{N_\tau}}) \ge \alpha\}. \quad (16)
$$

For any $\beta \in [0, \alpha]$, we derive that

$$
\mathbb{P}(S_{ut}^{N_\tau} \in [\mathcal{Q}^{N_\tau}(\beta), \mathcal{Q}^{N_\tau}(\beta+1-\alpha)] | \boldsymbol{S_1^{N_\tau}}) = 1 - \alpha. \quad (17)
$$

According to Eq.(6), we define the inverse of the score function as $\mu(v) = s^{-1}(v)$. For any $\alpha \in [0, 1]$, we have

$$
y_\alpha = f(X_{ut}^{N_\tau}) + \mu(\mathcal{Q}^{N_\tau}(\alpha)). \quad (18)
$$

Then the oracle prediction interval in continual learning with the narrowest width is defined as

$$
C_{ut}^\alpha = [y_{\hat\beta}, y_{\hat\beta+1-\alpha}]. \quad (19)
$$

$$
\hat\beta = \operatorname*{argmin}_{\beta \in [0,\alpha]} (\mu(\mathcal{Q}^{N_\tau}(\beta+1-\alpha)) - \mu(\mathcal{Q}^{N_\tau}(\beta))). \quad (20)
$$

For convenience, above discussion directly assumes that the score function is the one designed by our work. The above analysis indicates that the estimated quantile $\hat{\mathcal{Q}}$ and model $\hat{f}$ determine how closely the estimated prediction interval approximates the oracle prediction interval. In continual learning, $\hat{f}$ is the output model whose uncertainty we aim to quantify. So we focus on analyzing the estimated conditional quantile presented in this paper, i.e. Eq.(10). In this work, we want to seek the asymptotic coverage guarantee of the prediction interval of CPCL, which is defined as

$$
\mathbb{P}(Y_{ut}^{N_\tau} \in \widehat{C}_{ut}^\alpha | X_{ut}^{N_\tau}, \boldsymbol{Z_1^{N_\tau}}) \to 1 - \alpha \quad (21)
$$

with $N_{cal} \to \infty$. Therefore, proving Eq.(21) is equivalent to prove that for any $\alpha \in [0, 1]$, we have

$$
\hat{\mathcal{Q}}^{N_\tau}(x;\alpha) \to \mathcal{Q}^{N_\tau}(\alpha) \quad as \quad N_{cal} \to \infty, \quad (22)
$$

where $\hat{\mathcal{Q}}^{N_\tau}(x;\alpha)$ is the estimated conditional quantile of CPCL and $\mathcal{Q}^{N_\tau}(\alpha)$ is the true conditional quantile. In Eq.(10), we have $x = X_{N_{cal}}^R$ which corresponds to the score sequence $\boldsymbol{S_1^{N_\tau}}$. In Section 5.2, we prove Eq.(22) for any $x \in \mathbb{B}$.

## 5.2. Asymptotic Coverage Guarantee

Here we prove that Eq.(22) holds for any $\alpha \in [0, 1]$ and $x \in \mathbb{B}$. As stated in (Xu & Xie, 2023b), it is impossible to prove the conditional coverage guarantee without further assumptions. We first provide some necessary theoretical assumptions and briefly explain them.

**Assumption 1.** *The true conditional CDF in Eq.(15) is Lipschitz continuous with parameter $L$, i.e. for any $x, x' \in \mathbb{B}$,*

$$
\sup_s |F(s|x) - F(s|x')| \le L\|x - x'\|_1, \quad (23)
$$

*where $x, x'$ correspond to the score sequence.*

**Assumption 2.** *The true conditional CDF $F(s|x)$ is continuous and strictly monotonously increasing in $s$, for any $x \in \mathbb{B}$.*

Assumptions 1 and 2 make assumptions on the conditional CDF. The conditional distribution function is assumed to be Lipschitz continuous in Assumption 1 and strictly monotonously increasing in Assumption 2. These assumptions are reasonably mild and don't require a particular parametric.

**Assumption 3.** *For any $x \in \mathbb{B}$, $p_i(x)$ in Eq.(8) satisfies that $p_i(x) = o(1)$.*

**Assumption 4.** *For any $x \in \mathbb{B}$, the rectangular subspace $R_{l(x,\zeta_k)} \subseteq (0,1)^{N_\tau}$ of leaf $l(x, \zeta_k)$ of tree $\zeta_k$ is defined by the intervals $I(x, m, \zeta_k) \subseteq (0,1)$, i.e. $R_{\ell(x,\zeta_k)} = \otimes_{m=1}^{N_\tau} I(x, m, \zeta_k)$, where $\otimes$ means direct sum. We assume that $\max_m |I(x, m, \zeta_k)| = o_p(1)$ for $N_{cal} \to \infty$. $o_p(1)$ indicates that $\max_m |I(x, m, \zeta_k)|$ converges in probability to zero.*

Assumptions 3 and 4 focus on the actual construction of trees. We refer to Meinshausen (2006) and provide two examples. Assumption 3 represents the case of Example 1. Assumption 4 indicates that the size of each interval is vanishing for large $N_{cal}$ and represents the case of Example 2.

**Example 1.** *The minimal number of observations in a node is growing for large $N_{cal}$, i.e., $1/\min_{l,\zeta_k} w_{\zeta_k}(l) = o(1), N_{cal} \to \infty$..*

**Example 2.** *This example consists of three situations. In situation 1, the proportion of observations in a node, relative to all observations, is vanishing for large $N_{cal}$ i.e., $\max_{l,\zeta_k} w_{\zeta_k}(l) = o(N_{cal}), N_{cal} \to \infty$. In situation 2, when finding a variable for a splitpoint, the probability that variable $m = 1, ..., N_\tau$ is chosen for the splitpoint is bounded from below for every node by a positive constant. In situation 3, if a node is split, the split is chosen so that each of the resulting sub-nodes contains at least a proportion $\gamma$ of the observations in the original node, for some $0 < \gamma \le 0.5$.*

Discussion of Examples 1 and 2 can be found in Appendix C. Based on Assumptions 1 - 4, we prove that for any $x \in \mathbb{B}$, the conditional CDF $\hat{F}(s|x)$ of CPCL converges in probability to the true conditional CDF $F(s|x)$ as $N_{cal} \to \infty$.

**Theorem 1.** *Under Assumptions 1, 2, 3 and 4, for any $s \in (0,1)$ and $x \in \mathbb{B}$, the conditional CDF $\hat{F}(s|x)$ of CPCL (Eq.(9)) converges in probability to the true conditional CDF $F(s|x)$ as $N_{cal} \to \infty$, i.e.*

$$|\hat{F}(s|x) - F(s|x)| \to_p 0 \quad N_{cal} \to \infty \quad (24)$$

*holds pointwise for any $s \in (0,1)$ and $x \in \mathbb{B}$.*

The proof of Theorem 1 can be found in Appendix D.

**Remark 1.** *Theorem 1 represents a crucial step toward establishing the asymptotic coverage guarantee of the prediction interval of CPCL. This theorem demonstrates the*

*consistency of CPCL, showing that the error between the approximation of conditional distribution by CPCL and the true conditional distribution converges uniformly in probability to zero as $N_{cal} \to \infty$. Consequently, CPCL serves as a consistent method for estimating conditional distributions.*

Based on Theorem 1, we prove that for any $\alpha \in [0,1]$, the estimated conditional quantile of CPCL converges to the true conditional quantile. Theorem 2 shows this theoretical result. We rewrite $\mathcal{Q}^{N_\tau}(\alpha)$ as $\mathcal{Q}^{N_\tau}(x; \alpha)$ for convenience.

**Theorem 2.** *Under the same conditions as Theorem 1, for any $\alpha \in [0,1]$ and $x \in \mathbb{B}$, the estimated conditional quantile $\hat{\mathcal{Q}}^{N_\tau}(x;\alpha)$ of CPCL converges to the true conditional quantile $\mathcal{Q}^{N_\tau}(x; \alpha)$ as $N_{cal} \to \infty$, i.e.*

$$\hat{\mathcal{Q}}^{N_\tau}(x;\alpha) \to \mathcal{Q}^{N_\tau}(x; \alpha) \quad N_{cal} \to \infty \quad (25)$$

*holds for any $\alpha \in [0,1]$ and $x \in \mathbb{B}$.*

The proof of Theorem 2 can be found in Appendix E.

**Remark 2.** *Theorem 2 is sufficient to establish the asymptotic coverage guarantee of the prediction interval of CPCL. Specifically, $Y_{ut}^{N_\tau} \in \widehat{C}_{ut}^\alpha | X_{ut}^{N_\tau}, \mathbf{Z}_1^{\mathbf{N_\tau}}$ is equivalent to $S_{ut}^{N_\tau} \in [\hat{\mathcal{Q}}^{N_\tau}(X'; \beta), \hat{\mathcal{Q}}^{N_\tau}(X'; \beta+1-\alpha)] | \mathbf{S}_1^{\mathbf{N_\tau}}$ where $X' = X_{ut}^{N_\tau}$. According to Theorem 2, we know that $\mathbb{P}(S_{ut}^{N_\tau} \in [\hat{\mathcal{Q}}^{N_\tau}(X'; \beta), \hat{\mathcal{Q}}^{N_\tau}(X'; \beta+1-\alpha)] | \mathbf{S}_1^{\mathbf{N_\tau}})$ converges to $\mathbb{P}(S_{ut}^{N_\tau} \in [\mathcal{Q}^{N_\tau}(X'; \beta), \mathcal{Q}^{N_\tau}(X'; \beta+1-\alpha)] | \mathbf{S}_1^{\mathbf{N_\tau}})$ as $N_{cal} \to \infty$. Based on Eq.(17), we then conclude that $\mathbb{P}(Y_{ut}^{N_\tau} \in \widehat{C}_{ut}^\alpha | X_{ut}^{N_\tau}, \mathbf{Z}_1^{\mathbf{N_\tau}}) \to 1 - \alpha$ as $N_{cal} \to \infty$. This establishes the asymptotic coverage guarantee of the prediction interval for CPCL*

## 6. Experiments

In this section, we use simulated and real-world data to verify the validity of the prediction interval of CPCL. We consider typical continual learning methods, including SI (Zenke et al., 2017), EWC (Kirkpatrick et al., 2017), MAS (Aljundi et al., 2018), DGR (Shin et al., 2017)) and Fine-tuning on real-world data to explore the effect of continual learning methods. More details of these continual learning methods are in Appendix A.

### 6.1. Simulation

We follow the work of Zou & Liu (2024) to generate simulated data. In our setup, we consider two regression tasks within the continual learning process, setting $\mathcal{X} = \mathbb{R}^d$, $\mathcal{Y} = \mathbb{R}$. Define the oracle linear predictor $G : \mathcal{X} \to \mathcal{Y}$ as $G(x) = \langle w^\star, x \rangle + b^\star$, where $w^\star \in \mathbb{R}^d$ and $b^\star \in \mathbb{R}$. We define the marginal distribution of $X$ for tasks $\tau_1$ and $\tau_2$ as

$$F_{1X} = \mathcal{N}(u_1, \sigma_x^2 I_d), \quad F_{2X} = \mathcal{N}(u_2, \sigma_x^2 I_d), \quad (26)$$

respectively, where $u_1, u_2 \in \mathbb{R}^d$ are the mean vectors, $\sigma_x^2 > 0$ is a scalar and $I_d \in \mathbb{R}^{d \times d}$ is the identity matrix

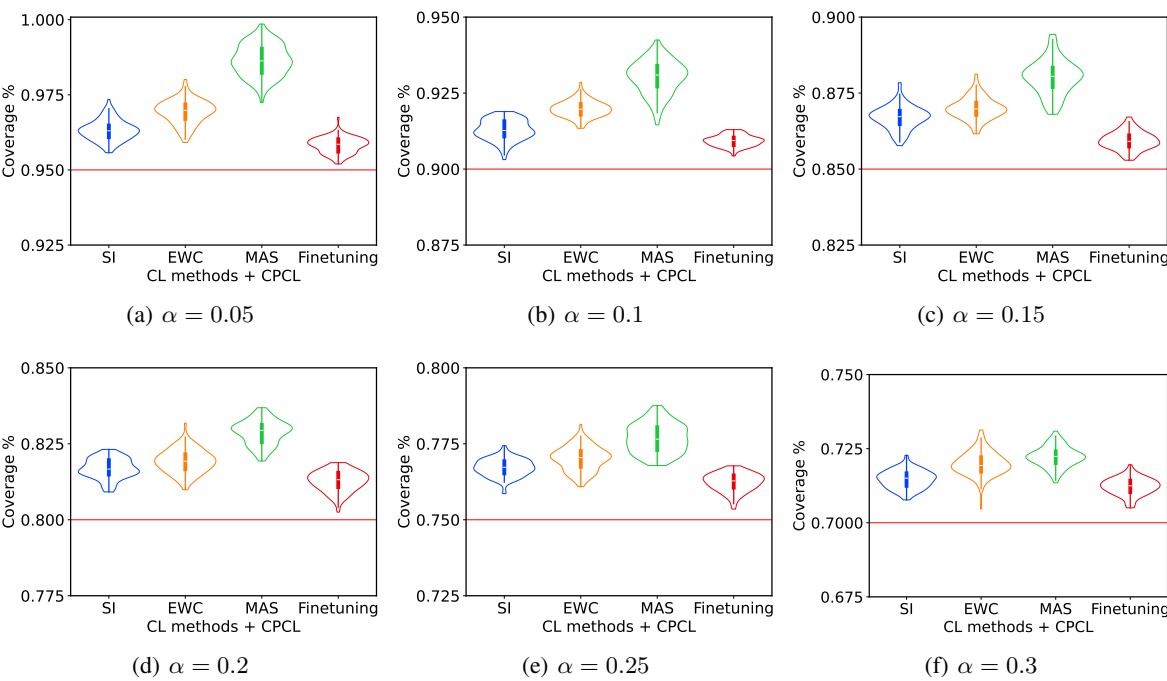

*Figure 2.* The violin plots illustrate the coverage for 100 experimental runs on the simulated data. The red lines indicate the marginal coverage guarantees we aim to achieve. In each violin, the white median line represents the median, while the endpoints of the thick line denote the 0.25 quantile and the 0.75 quantile. We present results for $\alpha = \{0.05, 0.1, 0.15, 0.2, 0.25, 0.3\}$.

of dimension $d \times d$. We define the conditional distribution $Y$ given $X$ as $F_Y = \mathcal{N}(G(x), \sigma_y^2)$ for both tasks $\tau_1$ and $\tau_2$. We draw 5000 samples from the above distribution to construct the train dataset for each task. We define the trained model as a linear predictor $\hat{G}(x) = \langle \hat{w}, x \rangle + \hat{b}$, where $\hat{w} \in \mathbb{R}^d$ and $\hat{b} \in \mathbb{R}$. We draw 1000 samples from the distribution of task $\tau_1$ and another 1000 samples from the distribution of task $\tau_2$ to construct the test dataset. We set $N_{cal} = 1000$ which is the number of samples for each task to construct the calibration set. After sequentially learning tasks $\tau_1$ and $\tau_2$, we quantify the uncertainty of the trained model $\hat{G}(x)$ using CPCL. The evaluation metric is the coverage which is the ratio between the number of test examples such that $y_i \in \widehat{C}_{ut}^\alpha(x_i)$ (Eq.(12)) and the size of the test dataset.

We set the significance level $\alpha$ to values in $\{0.05, 0.1, 0.15, 0.2, 0.25, 0.3\}$ and conduct 100 runs with different random seeds. Figure 2 shows the results of experiments on simulated data using CPCL. From the figure, we observe that the violins for CPCL with different continual learning methods are over the desired coverage lines. For example, the coverage results evaluated by CPCL exceed 0.95% when we use EWC as the continual learning method and $\alpha = 0.05$. These experiment results demonstrate the validity of our proposed CPCL on simulated data. We further study the effect of continual learning method with more tasks using real-world data.

### 6.2. Real-world Data

We conduct experiments using Tiny ImageNet, a subset of 200 classes from ImageNet (Deng et al., 2009), rescaled to an image size of $64 \times 64$. We perform 20 runs with different random seeds, randomly selecting 5 classes to form 5 tasks each time. In this case, there are 5 regression tasks in continual learning process. For each task, we have 500 samples belonging to one class, subdivided into training (80%) and calibration sets (20%) along with 50 samples for testing. We utilize a pretrained AlexNet (Krizhevsky et al., 2012) (denoted as $AN(\cdot)$, modified by replacing final layer with a linear layer suitable for regression) to define conditional distribution $Y$ given $X$ as $F_Y = \mathcal{N}(AN(x), \sigma_y^2)$. The trained model here is ResNet-18 (He et al., 2016) which is also modified for regression. After sequentially learning $T$ tasks, we quantify the uncertainty of the trained model by CPCL. We evaluate the results by calculating the coverage, utilizing testing samples from all $T$ previous tasks, as described in Section 6.1. While calculating the coverage, we also collect the length of each prediction interval and calculate the average prediction interval length over all seen test samples.

We set the significance level $\alpha$ to values in $\{0.1, 0.2, 0.3\}$. Figure 3 shows the results of experiments on real-world data using CPCL. From Figures 3(a), 3(b) and 3(c), we observe that the most swarms with different continual learning methods are over the desired coverage lines. These

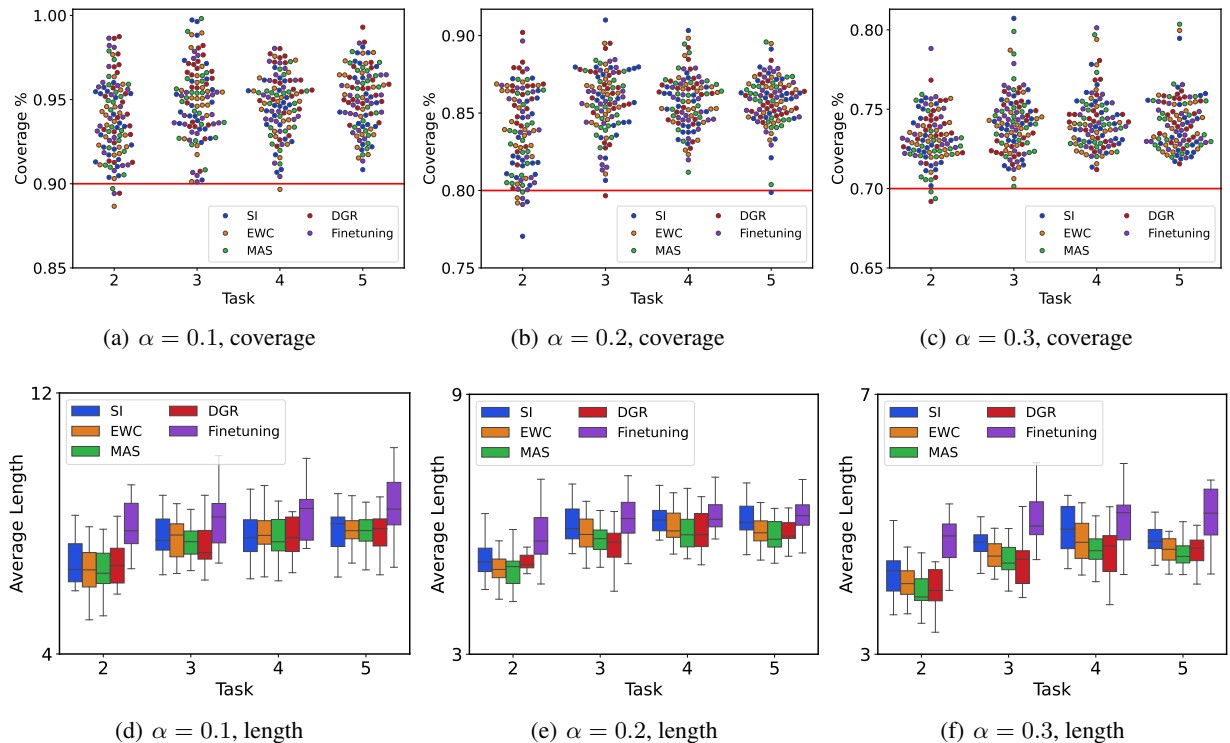

*Figure 3.* The swarm and box plots illustrate the coverage and average interval length from the 20 experimental runs on real-world data. In Figures 3(a), 3(b) and 3(c), the red lines indicate the marginal coverage guarantees we aim to achieve. In Figures 3(d), 3(e) and 3(f), the median line represents the median, while the two edges of the box correspond to the 0.25 quantile and the 0.75 quantile. Different colors represent different continual learning methods. We present results for $\alpha = \{0.1, 0.2, 0.3\}$.

demonstrate the validity of our proposed CPCL on real-world data. Figures 3(d), 3(e) and 3(f) show the average interval length. As the number of learning tasks increases, we find that the average interval length based on any continual learning method tends to increase. For example, the average length of the prediction interval can reach 9 after learning task 5, while it remains below 8 after learning task 2, when we us EWC as the continual learning method and $\alpha = 0.1$. This phenomenon results from catastrophic forgetting. Although continual learning methods are effective in mitigating catastrophic forgetting, some forgetting still occurs as the number of learning tasks increases. Eq.(12) shows that the prediction interval is based on the resulted model $\hat{f}_T$. Consequently, an increasing prediction interval length indicates model forgetting. Therefore, our proposed CPCL effectively reflects the phenomenon of forgetting and measures the performance of continual learning methods.

## 7. Conclusion

In this paper, we propose a **CP**-based method for model uncertainty quantification in **c**ontinual **l**earning, termed **CPCL**. CPCL collects the calibration set through replay and designs a nonconformity score function to construct a score

set, where scores are sequentially dependent. It accounts for these dependencies by reconstructing a new dataset based on score set. CPCL trains QRF using reconstructed dataset and defines the prediction interval by the conditional quantile estimator derived from the trained QRF. Then we analyze the oracle prediction interval in continual learning and provide the asymptotic coverage guarantee for the prediction interval of CPCL. Finally, extensive experiments on simulated and real data empirically verify the validity of CPCL.

## Acknowledgements

This work is supported by the Key R&D Program of Hubei Province under Grant 2024BAB038, the National Key R&D Program of China under Grant 2023YFC3604702, the Fundamental Research Funds for the Central Universities under Grant 2042025kf0045.

## Impact Statement

This paper presents work whose goal is to advance the field of Machine Learning. There are many potential societal consequences of our work, none which we feel must be specifically highlighted here.

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

# A. Additional Experimental Setting

Here, we provide more information about continual learning methods. There are five methods in our paper. We give some descriptions of these baselines.

**SI:** The synaptic state tracks the past and current parameter value, and maintains an online estimate of the synapse's importance toward solving problems encountered in the past.

**MAS:** Redefines the parameter importance measure to an unsupervised setting and obtains gradients of the squared $L_2$-norm of the learned network output function.

**EWC:** EWC introduces network parameter uncertainty in the Bayesian framework; the true posterior is estimated using a Laplace approximation with precision, determined by the Fisher Information Matrix (FIM), which shows equivalence to the positive semi-definite second order derivative of the loss near a minimum.

**DGR:** DGR consists of a deep generative model ("generator") and a task solving model ("solver"). DGR trains a deep generative model in the generative adversarial networks (GANs) framework to mimic previous data. Generated data are then paired with corresponding response from the previous task solver to represent old tasks. Called the scholar model, the generator-solver pair can produce fake data and desired target pairs as much as needed, and when presented with a new task, these produced pairs are interleaved with new data to update the generator and solver networks.

**Finetuning:** Finetuning greedily trains each task without considering previous task performance—hence introducing catastrophic forgetting—and represents the minimum desired performance.

# B. Discussion of Nonconformity Score Function

We introduce the sigmoid-based nonconformity score function for two reasons. (1) The range of sigmoid-based nonconformity score function is $(0, 1)$. We train QRF on the reconstructed dataset $D_T^R = \{(X_i^R, Y_i^R)\}_{i=1}^{N_{cal}-1}$, where each entry $S_i^t$ in $X_i^R$ is the calculated score. The training of QRF requires to construct trees. This process involves the intervals $I(x, m, \zeta_k) \subseteq (0, 1)$. The interval $I(x, m, \zeta_k)$ is used to determine whether $X_i^R$ is in the node corresponding to $I(x, m, \zeta_k)$, i.e., if $S_i^m \in I(x, m, \zeta_k)$, then $X_i^R$ is in the corresponding node. Since $I(x, m, \zeta_k) \subseteq (0, 1)$, we need to define that the score function ranges in $(0, 1)$ for rigorous proof. (2) Sigmoid-based nonconformity score function is invertible. In Eq.(11) we present the prediction interval in terms of scores. The invertible function helps us to rewrite the prediction interval which is shown in Eq.(12). A typical score function in CP is $s(X_i^t, Y_i^t) = |\hat{\mu}_i^t|$ where $\hat{\mu}_i^t = Y_i^t - \hat{f}_T(X_i^t)$. It is not invertible and the corresponding range is not $(0, 1)$. Therefore, it is not suitable in this paper.

# C. Examples

We refer to Meinshausen (2006) and provide two examples. Assumption 3 represents the case of Example 1. Assumption 4 indicates that the size of each interval is vanishing for large $N_{cal}$ and represents the case of Example 2.

For each tree with separate parameter $\zeta_k$, there are $L$ leaves, where every leaf $l$ is associated with a rectangular subspace $R_l \subset \mathbb{B}$. These subspaces are disjoint and cover the entire space $\mathbb{B}$, i,.e. for any $x \in \mathbb{B}$, there is one and only one leaf which corresponds to $R_{l(x, \zeta_k)}$. Denote the node-sizes of the leaves $l$ of a tree by $w_{\zeta_k}(l) = \|$
$i \in \{1, \ldots, N_{cal} - 1\} : X_i \in R_{l(x, \zeta_k)}\|$. $X_i \in (0, 1)^{N_\tau}$ is the observation to train QRF.

We discuss Assumption 3 by Example 1: The minimal number of observations in a node is growing for large $N_{cal}$, i.e., $1/\min_{l, \zeta_k} w_{\zeta_k}(l) = o(1), N_{cal} \to \infty.$. Recalling $p_i(x) = \sum_{k=1}^{K} p_i(x, \zeta_k)/K$, we have $0 \le p_i(x) \le 1/\min_{l, \zeta_k} w_{\zeta_k}(l) = o(1)$, which means that $p_i(x) = o(1)$ for any $x \in \mathbb{B}$. Therefore, Assumption 3 represents the case of Example 1.

We discuss Assumption 4 by Example 2 which consists of three situations. In situation 1, the proportion of observations in a node, relative to all observations, is vanishing for large $N_{cal}$ i.e., $\max_{l, \zeta_k} w_{\zeta_k}(l) = o(N_{cal}), N_{cal} \to \infty$. In situation 2, when finding a variable for a splitpoint, the probability that variable $m = 1, ..., N_\tau$ is chosen for the splitpoint is bounded from below for every node by a positive constant. In situation 3, if a node is split, the split is chosen so that each of the resulting sub-nodes contains at least a proportion $\gamma$ of the observations in the original node, for some $0 < \gamma \le 0.5$.

As any $x \in \mathbb{B}$ is dropped down a tree, several nodes are passed. Denote by $S(x, m, \zeta_k)$ the number of times that these nodes contain a splitpoint on variable $m$. The total number of nodes that $x$ passes through is denoted by $S(x, \zeta_k) =$

$\sum_{m=1}^{N_\tau} S(x, m, \zeta_k)$. Using situation 3, the maximal number of observations in any leaf, $\max_l w_{\zeta_k}(l)$ is bounded (for every tree) from below by $N_{cal}\gamma^{S_{\min}(\zeta_k)}$ where $S_{\min}(\zeta_k) = \min_{x\in\mathbb{B}} S(x, \zeta_k)$. Using situation 1, $\max_l w_{\zeta_k}(l)$ is on the other hand bounded from above by an $o(N_{cal})$-term. Putting together, we conclude that $\gamma^{S_{\min}(\zeta_k)} = o(1)$ for $N_{cal} \to \infty$. Hence there exists a sequence $s_{N_{cal}}$ with $s_{N_{cal}} \to \infty$ for $N_{cal} \to \infty$ such that $S_{\min}(\zeta_k) \geq s_{N_{cal}}$. As the probability of splitting on variable $m = 1, ..., N_\tau$ is bounded from below by a positive constant, by situation 2, there exists a sequence $g_{N_{cal}}$ with $g_{N_{cal}} \to \infty$ for $N_{cal} \to \infty$ such that $P\{\min_m S(x, m, \zeta_k) > g_{N_{cal}}\} \to 1 \quad N_{cal} \to \infty$. Using situation 3, we obtain that $|\{i \in \{1, \ldots, N_{cal} - 1\} : X_{i,m} \in I(x, m, \zeta_k)\}|/(N_{cal} - 1) \leq (1 - \gamma)^{S(x,m,\zeta_k)}$. Putting together, we conclude that $\max_m |\{i \in \{1, ..., N_{cal} - 1\} : X_{i,m} \in I(x, m, \zeta_k)\}|/(N_{cal} - 1) = o_p(1)$ which indicates $\max_m |I(x, m, \zeta_k)| = o_p(1)$ for $N_{cal} \to \infty$. Therefore, Assumption 4 represents the case of Example 2.

## D. Proof of Theorem 1

Theorem 1 shows that the conditional CDF $\hat{F}(s|x)$ of CPCL converges in probability to the true conditional CDF $F(s|x)$ as $N_{cal} \to \infty$. We recall Theorem 1 and prove it.

**Theorem 1.** *Under Assumptions 1, 2, 3 and 4, for any $s \in (0, 1)$ and $x \in \mathbb{B}$, the conditional CDF $\hat{F}(s|x)$ of CPCL converges in probability to the true conditional CDF $F(s|x)$ as $N_{cal} \to \infty$, i.e.*

$$|\hat{F}(s|x) - F(s|x)| \to_p 0 \quad N_{cal} \to \infty \tag{D.1}$$

*holds pointwise for any $s \in (0, 1)$ and $x \in \mathbb{B}$.*

*Proof.* For each observation in $\{(X_i^R, Y_i^R)\}_{i=1}^{N_{cal}-1}$, we define

$$U_i = F(Y_i^R|X_i^R) \tag{D.2}$$

as the quantile of the $i$-th empirical score $Y_i^R$ (corresponding to $s$ in CDF). We replace $Y_i^R$ with $S_i^R$ for convenience in this proof. It is noteworthy that $U_i \sim \text{Unif}[0, 1]$ due to the assumption of continuous distribution function in Assumption 2. We then derive the point-wise difference between the conditional CDF $\hat{F}(s|x)$ of CPCL and the true conditional CDF $F(s|x)$.

$$\hat{F}(s|x) - F(s|x) = \sum_{i=1}^{N_{cal}-1} p_i(x)\mathbb{I}(S_i^R \leq s) - F(s|x) \tag{D.3}$$

Under Assumption 2, we know that $\{S_i^R \leq s\}$ is identical to $\{U_i \leq F(s|X_i^R)\}$. Therefore, we derive that

$$\hat{F}(s|x) - F(s|x) = \sum_{i=1}^{N_{cal}-1} p_i(x)\mathbb{I}(U_i \leq F(s|X_i^R)) - F(s|x)$$

$$= \sum_{i=1}^{N_{cal}-1} p_i(x)\mathbb{I}(U_i \leq F(s|x)) - F(s|x) + \sum_{i=1}^{N_{cal}-1} p_i(x)(\mathbb{I}(U_i \leq F(s|X_i^R)) - \mathbb{I}(U_i \leq F(s|x)))$$

$$\tag{D.4}$$

Taking the absolute value of both sides of Eq.(D.4), we derive that

$$|\hat{F}(s|x) - F(s|x)| \leq |\sum_{i=1}^{N_{cal}-1} p_i(x)\mathbb{I}(U_i \leq F(s|x)) - F(s|x)| + |\sum_{i=1}^{N_{cal}-1} p_i(x)(\mathbb{I}(U_i \leq F(s|X_i^R)) - \mathbb{I}(U_i \leq F(s|x)))|$$

$$\tag{D.5}$$

The first term of right side of Eq.(D.5) is a variance-type part, while the second term of that reflects the change in the underlying distribution. For the first term, we derive the expectation:

$$\mathbb{E}[\sum_{i=1}^{N_{cal}-1} p_i(x)\mathbb{I}(U_i \leq F(s|x))] = \sum_{i=1}^{N_{cal}-1} p_i(x)\mathbb{E}[\mathbb{I}(U_i \leq F(s|x))] \tag{D.6}$$

Due to $U_i \sim \text{Unif}[0, 1]$, we conclude that $\mathbb{E}[\mathbb{I}(U_i \leq F(s|x))] = F(s|x))$. According to Eq.(8) of Section 4, we know that $\sum_{i=1}^{N_{cal}-1} p_i(x) = 1$. Therefore, we derive that

$$\mathbb{E}[\sum_{i=1}^{N_{cal}-1} p_i(x)\mathbb{I}(U_i \leq F(s|x))] = \sum_{i=1}^{N_{cal}-1} p_i(x)F(s|x) = F(s|x) \tag{D.7}$$

Based on Eq.(D.7), we rewrite the first term of right side of Eq.(D.5) as

$$| \sum_{i=1}^{N_{cal}-1} p_i(x)\mathbb{I}(U_i \leq F(s|x)) - F(s|x)| = | \sum_{i=1}^{N_{cal}-1} p_i(x)\mathbb{I}(U_i \leq F(s|x)) - \mathbb{E}[\sum_{i=1}^{N_{cal}-1} p_i(x)\mathbb{I}(U_i \leq F(s|x))]| \qquad (D.8)$$

By Chebyshev inequality, for any $\epsilon > 0$, we have

$$\mathbb{P}(| \sum_{i=1}^{N_{cal}-1} p_i(x)\mathbb{I}(U_i \leq F(s|x)) - F(s|x)| \geq \epsilon)$$

$$= \mathbb{P}(| \sum_{i=1}^{N_{cal}-1} p_i(x)\mathbb{I}(U_i \leq F(s|x)) - \mathbb{E}[\sum_{i=1}^{N_{cal}-1} p_i(x)\mathbb{I}(U_i \leq F(s|x))]| \geq \epsilon) \qquad (D.9)$$

$$\leq Var(\sum_{i=1}^{N_{cal}-1} p_i(x)\mathbb{I}(U_i \leq F(s|x)))/\epsilon^2$$

Due to the way of constructing $\{(X_i^R, Y_i^R)\}_{i=1}^{N_{cal}-1}$, we let $U_i$ for $i = 1, \ldots, N_{cal} - 1$ independent. Therefore, we have

$$Var(\sum_{i=1}^{N_{cal}-1} p_i(x)\mathbb{I}(U_i \leq F(s|x)))$$

$$= \sum_{i=1}^{N_{cal}-1} p_i(x)^2 Var(\mathbb{I}(U_i \leq F(s|x)))$$

$$= \sum_{i=1}^{N_{cal}-1} p_i(x)^2 (\mathbb{E}[\mathbb{I}(U_i \leq F(s|x))] - \mathbb{E}[\mathbb{I}(U_i \leq F(s|x))]^2) \qquad (D.10)$$

$$= \sum_{i=1}^{N_{cal}-1} p_i(x)^2 (F(s|x) - F^2(s|x))$$

$$< \sum_{i=1}^{N_{cal}-1} p_i(x)^2.$$

By Assumption 3 and Eq.(8), we have $0 < p_i(x) = o(1)$ and $\sum_{i=1}^{N_{cal}-1} p_i(x) = 1$. Therefore, for any $x \in \mathbb{B}$, we derive that

$$\sum_{i=1}^{N_{cal}-1} p_i(x)^2 \to 0 \quad N_{cal} \to \infty. \qquad (D.11)$$

Combining Eqs.(D.9), (D.10) and (D.11), we derive that the first term of right side of Eq.(D.5) converges to 0 as $N_{cal} \to \infty$, i.e.

$$| \sum_{i=1}^{N_{cal}-1} p_i(x)\mathbb{I}(U_i \leq F(s|x)) - F(s|x)| \to_p 0 \quad N_{cal} \to \infty. \qquad (D.12)$$

holds for any $s \in (0,1)$ and $x \in \mathbb{B}$. Then we turns attention to the second term of right side of Eq.(D.5). We define $H(U_i)$ as

$$H(U_i) = \sum_{i=1}^{N_{cal}-1} p_i(x)(\mathbb{I}(U_i \leq F(s|X_i^R)) - \mathbb{I}(U_i \leq F(s|x))). \qquad (D.13)$$

Due to all $U_i$ are uniform over [0,1], it holds that

$$\mathbb{E}[H(U_i)] = \mathbb{E}[\sum_{i=1}^{N_{cal}-1} p_i(x)(\mathbb{I}(U_i \leq F(s|X_i^R)) - \mathbb{I}(U_i \leq F(s|x)))]$$

$$= \sum_{i=1}^{N_{cal}-1} p_i(x)\mathbb{E}[\mathbb{I}(U_i \leq F(s|X_i^R)) - \mathbb{I}(U_i \leq F(s|x))] \tag{D.14}$$

$$= \sum_{i=1}^{N_{cal}-1} p_i(x)[F(s|X_i^R) - F(s|x)]$$

By Chebyshev inequality, for any $\epsilon > 0$, we have

$$\mathbb{P}(|H(U_i) - \mathbb{E}[H(U_i)]| \geq \epsilon) \leq Var(H(U_i))/\epsilon^2. \tag{D.15}$$

Similar to the proof process of Eq.(D.10), we derive that

$$Var(H(U_i)) = Var(\sum_{i=1}^{N_{cal}-1} p_i(x)(\mathbb{I}(U_i \leq F(s|X_i^R)) - \mathbb{I}(U_i \leq F(s|x))))$$

$$= \sum_{i=1}^{N_{cal}-1} p_i(x)^2 Var(\mathbb{I}(U_i \leq F(s|X_i^R)) - \mathbb{I}(U_i \leq F(s|x)))$$

$$= \sum_{i=1}^{N_{cal}-1} p_i(x)^2(\mathbb{E}[\mathbb{I}(U_i \leq F(s|X_i^R)) - \mathbb{I}(U_i \leq F(s|x))] - \mathbb{E}[\mathbb{I}(U_i \leq F(s|X_i^R)) - \mathbb{I}(U_i \leq F(s|x))]^2)$$

$$= \sum_{i=1}^{N_{cal}-1} p_i(x)^2(F(s|X_i^R) - F(s|x) - (F(s|X_i^R) - F(s|x))^2)$$

$$\leq \sum_{i=1}^{N_{cal}-1} p_i(x)^2$$

$$\tag{D.16}$$

Combining Eqs.(D.11) and (D.16), we derive that $H(U_i) \to_p \mathbb{E}[H(U_i)]$ as $N_{cal} \to \infty$. Therefore, we have

$$|\sum_{i=1}^{N_{cal}-1} p_i(x)(\mathbb{I}(U_i \leq F(s|X_i^R)) - \mathbb{I}(U_i \leq F(s|x)))| \to_p |\sum_{i=1}^{N_{cal}-1} p_i(x)(F(s|X_i^R) - F(s|x))| \quad as \quad N_{cal} \to \infty.$$

$$\tag{D.17}$$

According to Assumption 1, we derive that

$$|\sum_{i=1}^{N_{cal}-1} p_i(x)(F(s|X_i^R) - F(s|x))| \leq \sum_{i=1}^{N_{cal}-1} p_i(x)|F(s|X_i^R) - F(s|x)|$$

$$\leq \sum_{i=1}^{N_{cal}-1} p_i(x)\sup_s |(F(s|X_i^R) - F(s|x))| \tag{D.18}$$

$$\leq \sum_{i=1}^{N_{cal}-1} p_i(x)L\|X_i^R - x\|_1$$

By Assumption 4, we have $\max_m |I(x, m, \zeta_k)| = o_p(1)$ for $N_{cal} \to \infty$ and any $x \in \mathbb{B}$, which suffices to show that

$$\sum_{i=1}^{N_{cal}-1} p_i(x)L\|X_i^R - x\|_1 = o_p(1). \tag{D.19}$$

We therefore conclude that the second term of right side of Eq.(D.5) converges to 0 as $N_{cal} \to \infty$. Overall, we prove that under Assumptions 1, 2, 3 and 4, for any $s \in (0,1)$ and $x \in \mathbb{B}$, the conditional CDF $\hat{F}(s|x)$ of CPCL converges in

probability to the true conditional CDF $F(s|x)$ as $N_{cal} \to \infty$, i.e.

$$|\hat{F}(s|x) - F(s|x)| \to_p 0 \quad N_{cal} \to \infty \tag{D.20}$$

holds pointwise for any $s \in (0, 1)$ and $x \in \mathbb{B}$. Xu & Xie (2023a) provide a similar proof which can also be referred to. $\square$

# E. Proof of Theorem 2

Theorem 2 shows that the estimated conditional quantile $\hat{\mathcal{Q}}^{N_\tau}(x;\alpha)$ of CPCL converges to the true conditional quantile $\mathcal{Q}^{N_\tau}(x;\alpha)$ as $N_{cal} \to \infty$. We recall Theorem 2 and prove it.

**Theorem 2.** *Under the same conditions as Theorem 1, for any $\alpha \in [0, 1]$ and $x \in \mathbb{B}$, the estimated conditional quantile $\hat{\mathcal{Q}}^{N_\tau}(x;\alpha)$ of CPCL converges to the true conditional quantile $\mathcal{Q}^{N_\tau}(x;\alpha)$ as $N_{cal} \to \infty$, i.e.*

$$\hat{\mathcal{Q}}^{N_\tau}(x;\alpha) \to \mathcal{Q}^{N_\tau}(x;\alpha) \quad N_{cal} \to \infty \tag{E.21}$$

*holds for any $\alpha \in [0, 1]$ and $x \in \mathbb{B}$.*

*Proof.* Proving Eq.(E.21) is identical to prove that

$$|\hat{\mathcal{Q}}^{N_\tau}(x;\alpha) - \mathcal{Q}^{N_\tau}(x;\alpha)| \to 0 \quad N_{cal} \to \infty \tag{E.22}$$

holds for any $x \in \mathbb{B}$ and $\alpha \in [0, 1]$. Recall that

$$\mathcal{Q}^{N_\tau}(x; \alpha) = \inf\{s : F(s|x) \geq \alpha\}. \tag{E.23}$$

By Assumption 2, we know that $F(s|x)$ is continuous and strictly monotonously increasing in $s$. We consider a small perturbation of $\mathcal{Q}^{N_\tau}(x; \alpha)$, which results in the changes of the conditional distribution function, i.e. $|F(\mathcal{Q}^{N_\tau}(x;\alpha) - \epsilon|x) - \alpha|$ and $|F(\mathcal{Q}^{N_\tau}(x;\alpha) + \epsilon|x) - \alpha|$. We define this changes as $\delta = \min(|F(\mathcal{Q}^{N_\tau}(x;\alpha) - \epsilon|x) - \alpha|, |F(\mathcal{Q}^{N_\tau}(x;\alpha) + \epsilon|x) - \alpha|)$. Therefore, we consider the estimated quantile $\hat{\mathcal{Q}}^{N_\tau}(x;\alpha)$ as the perturbation of $\mathcal{Q}^{N_\tau}(x;\alpha)$. When the $|\hat{\mathcal{Q}}^{N_\tau}(x;\alpha) - \mathcal{Q}^{N_\tau}(x;\alpha)| > \epsilon$, the change of $F(s|x)$ is more than $\delta$, i.e. $|F(\hat{\mathcal{Q}}^{N_\tau}(x;\alpha) - \alpha| > \delta$. Then we derive that

$$\begin{aligned} \mathbb{P}(|\hat{\mathcal{Q}}^{N_\tau}(x;\alpha) - \mathcal{Q}^{N_\tau}(x;\alpha)| > \epsilon) &= \mathbb{P}(|F(\hat{\mathcal{Q}}^{N_\tau}(x;\alpha) - \alpha| > \delta) \\ &= \mathbb{P}(|F(\hat{\mathcal{Q}}^{N_\tau}(x;\alpha) - \hat{F}(\hat{\mathcal{Q}}^{N_\tau}(x;\alpha)| > \delta) \end{aligned} \tag{E.24}$$

According to the consistency in Theorem 1, we have

$$\mathbb{P}(|F(\hat{\mathcal{Q}}^{N_\tau}(x;\alpha) - \hat{F}(\hat{\mathcal{Q}}^{N_\tau}(x;\alpha)| > \delta) \to 0 \quad as \quad N_{cal} \to \infty. \tag{E.25}$$

Combining Eqs.(E.24) and (E.25), we derive that

$$\mathbb{P}(|\hat{\mathcal{Q}}^{N_\tau}(x;\alpha) - \mathcal{Q}^{N_\tau}(x;\alpha)| > \epsilon) \to 0 \quad as \quad N_{cal} \to \infty. \tag{E.26}$$

This means that

$$|\hat{\mathcal{Q}}^{N_\tau}(x;\alpha) - \mathcal{Q}^{N_\tau}(x;\alpha)| \to 0 \quad as \quad N_{cal} \to \infty. \tag{E.27}$$

Overall, we prove that the estimated conditional quantile $\hat{\mathcal{Q}}^{N_\tau}(x;\alpha)$ of CPCL converges to the true conditional quantile $\mathcal{Q}^{N_\tau}(x;\alpha)$ as $N_{cal} \to \infty$.

$\square$

