# OpenReview forum: "Model Uncertainty Quantification by Conformal Prediction in Continual Learning"
_ICML.cc/2025/Conference — ICML 2025 poster_

### Official Review · Reviewer_kvQq · 2025-03-11

**Overall Recommendation:** 4

**Summary:**

The paper addresses the problem of continual learning with calibration guarantees. More precisely, the purpose is to train a model to address a series of tasks, in a sequential way (i.e., one task after the other). The datasets used to train the model on the successive tasks are not exchangeable, and may even be forgotten. Then, building a calibration set becomes challenging. In a nutshell, to overcome this issue, the authors propose to use a replay-based strategy to construct the calibration dataset; this dataset is used to train a quantile regression model, based on which nonconformity scores can be predicted: the dataset is first "reconstructed" (features being obtained from the past nonconformity scores and outputs from the average current nonconformity scores). Thus, for any test instance based on the forest, nonconformity scores can be obtained, and therefore prediction intervals.

The introduction recalls the setting, motivates the paper and provides the main ideas of the contribution. Section 2 presents the related works, starting with continual learning and proceeding with a short refresher on conformal prediction. Section 3 provides a formalization of the problem addressed. Section 4 presents the main contribution of the paper, explaining how the calibration set is obtained, presenting the considered nonconformity score, and explaining how the quantile regression forest can be trained to predict NC scores and how decisions can subsequently be made. The whole pipeline is summarized in an algorithm. Section 5 provides a theoretical analysis of the proposal, providing a consistency property based on four assumptions. Section 6 reports the experiments realized on a synthetic dataset and on a real dataset (Tiny ImageNet). Section 7 concludes the paper.

### update after rebuttal

I would like to thank the authors for their aswers to my comments and questions. I updated my score accordingly.

**Claims And Evidence:**

The paper presents a strategy for continual learning with calibration guarantees. This claim is backed by a theoretical analysis, and more precisely formalized into two theorems. The experiments realized are rather simple (a synthetic dataset, and a real-world dataset), but nevertheless back up the proposal.

Overall, the evidence provided rather convincingly supports the claim, even if the strength of the assumptions in Section 5 is not discussed. Some components in the proposal are also rapidly presented, and could have been better justified or at least clarified.

**Essential References Not Discussed:**

N/A

**Experimental Designs Or Analyses:**

The experimental study conducted seems valid; the experimental setup is convincing and the results are as the reader could expect.

**Methods And Evaluation Criteria:**

The experimental setting and the datasets considered make sense for the problem at hand, even if a more thorough experimental analysis would have been appreciated.

**Other Comments Or Suggestions:**

I'd suggest to improve the writing.

Some parts are difficult to understand, e.g. the end of the "Nonconformity score function" paragraph in Section 4 (page 4). As well, the paragraph dedicated to quantile regression forests (mainly the part page 5) is difficult to follow due to the notations, to the lack of explanations, and to an extensive use of math in the text.

In Equation (1), $\mu$ would have been formally defined, and the assumptions pertaining to it could have been made explicit. It seems that a norm is missing either in Equation (5) or in Equation (6). Equation (13) seems to be disconnected from what precedes.

There are some (minor) typos. A couple of words are missing here and there, e.g. "In continual learning setting" (page 2), "the scores in score set" (page 4), "score set", "on calibration set", "that conditional distribution function" (page 5). Page 2, in "(IMM) (Lee et al., 2017)", there should not be parentheses around "IMM". In Section 3 (page 4), $Z_{ut}=(X_{ut},Y_{ut})$ is not properly defined.

Some references seem incomplete, e.g. "Krizhevsky, A., Sutskever, I., and Hinton, G. E. Imagenet
classification with deep convolutional neural networks. In NeurIPS, 2012."

**Other Strengths And Weaknesses:**

The paper is overall well written, but sometimes difficult to follow (see below for some comments).

**Questions For Authors:**

Could you elaborate on the nonconformity score function such as defined by Equation (6) ?

Could you discuss the assumptions made to establish the consistency results in Section 5.2 ?

**Relation To Broader Scientific Literature:**

The discussion on the related works seems to include the main references pertaining to the problem addressed.

**Theoretical Claims:**

I only briefly checked the soundness of the proofs (provided in the appendices); they seem correct.

---

> ### Author Rebuttal · Authors · 2025-04-01
>
> **R to OCOS1.** The end of the "Nonconformity score function" paragraph in Section 4 mainly discusses the score set calculated by the nonconformity score function. We are sorry that our notations is hard to follow. We will revised the confusing notations.
>
> **R to OCOS2.** We will improve our writing of all equations in our paper to makes them easy to read. No norm is missing in Equations (5) and (6). Equation (13) aims to calculate the $\hat{\beta}$ in Equation (12).
>
> **R to OCOS3 and OCOS4 .** We will fix these typos and incomplete references.
>
> **R to Q1.** We introduce the sigmoid-based nonconformity score function for two reasons. Please refer to  **R to Q1.** of Reviewer  vzZS.
>
> **R to Q2.**  Assumptions 1 and 2 make assumptions on the conditional CDF. The conditional distribution function is assumed to be Lipschitz continuous in Assumption 1 and strictly monotonously increasing in Assumption 2. Assumptions 3 and 4 focus on the actual construction of trees. For each tree with separate parameter $\zeta _{k}$, there are $L$ leaves, where every leaf $l$ is associated with a rectangular subspace $R _{l}\subset\mathbb{B}$. These subspaces are disjoint and cover the entire space $\mathbb{B}$, i,.e. for any $x\in\mathbb{B}$, there is one and only one leaf which corresponds to $R _{l(x,\zeta _{k})}$. Denote the node-sizes of the leaves $l$  of a tree by $w _{\zeta _{k}}(l)=\|\\{i\in\{1,\ldots,N _{cal}-1\}:X _{i}\in R _{l(x,\zeta _{k})}\\}\|$. $X _{i}\in(0,1)^{N _{\tau}}$ is the observation to train QRF.
>
> The key of Assumptions 3 is "For any $x\in \mathbb{B}$, $p _{i}(x)$ satisfies that $p _{i}(x)=o(1)$". We discuss Assumption 3 by  **Example 1**: The minimal number of observations in a node is growing for large $N _{cal}$, i.e., $1/\operatorname*{min} _{l,\zeta _{k}} w _{\zeta _{k}}(l) =o(1), N _{cal}\to\infty.$. Recalling $p _{i}(x)=\sum _{k=1}^{K} p _{i}(x,\zeta _{k})/K$, we have $0\leq p _{i}(x)\leq 1/\operatorname*{min} _{l,\zeta _{k}}w _{\zeta _{k}}(l) =o(1)$, which means that  $p _{i}(x)=o(1)$ for any $x\in \mathbb{B}$. Therefore, Assumption 3 represents the case of **Example 1**.
>
> Assumptions 4 is "For any $x\in \mathbb{B}$, the rectangular subspace $R_{l(x,\zeta_{k})}\subseteq(0,1)^{N_{\tau}}$ of leaf $l(x,\zeta_{k})$ of tree $\zeta_{k}$ is defined by the intervals $I(x,m,\zeta_{k})\subseteq(0,1)$, i.e. $R_{\ell(x,\zeta_{k})}=\otimes_{m=1}^{N_{\tau}}I(x,m,\zeta_{k})$, where $\otimes$ means direct sum. We assume that $\max_{m}|I(x,m,\zeta_{k})|=o_{p}(1)$ for $N_{cal}\to \infty$". We discuss Assumption 4 by  **Example 2** which consists of three situations. In **situation 1**, the proportion of observations in a node, relative to all observations, is vanishing for large $N _{cal}$ i.e., $\operatorname*{max} _{l,\zeta _{k}} w _{\zeta _{k}}(l) =o(N _{cal}), N _{cal}\to\infty.$.  In **situation 2**, when finding a variable for a splitpoint, the probability that variable $m=1,...,N _{\tau}$ is chosen for the splitpoint is bounded from below for every node by a positive constant. In **situation 3**, if a node is split, the split is chosen so that each of the resulting sub-nodes contains at least a proportion $\gamma$ of the observations in the original node, for some $0<\gamma\leq0.5 $.
>
> As any $x\in \mathbb{B}$ is dropped down a tree, several nodes are passed. Denote by $S(x,m,\zeta _{k})$ the number of times that these nodes contain a splitpoint on variable $m$. The total number of nodes that $x$ passes through is denoted by $S(x,\zeta _{k})=\sum _{m=1}^{N _{\tau}} S(x,m,\zeta _{k}).$ Using **situation 3**, the maximal number of observations in any leaf, $\operatorname*{max} _{l} w _{\zeta _{k}}(l)$ is bounded (for every tree) from below by $N _{cal} \gamma^{S _{\min}(\zeta _{k})}$ where $S _{\min}(\zeta _{k})=\operatorname*{min} _{x\in  \mathbb{B}} S(x,\zeta _{k})$. Using **situation 1**, $\operatorname*{max} _{l} w _{\zeta _{k}}(l)$ is on the other hand bounded from above by an $o(N _{cal})$-term. Putting together, we conclude that $\gamma^{S _{\min}(\zeta _{k})}=o(1)$ for  $N _{cal}\to \infty$. Hence there exists a sequence $s _{N _{cal}}$ with $s _{N _{cal}} \to \infty$ for $N _{cal}\to \infty$ such that $S _{\min}(\zeta _{k}) \geq s _{N _{cal}}$. As the probability of splitting on variable $m=1,...,N _{\tau}$ is bounded from below by a positive constant, by **situation 2**, there exists a sequence $g _{N _{cal}}$ with $g _{N _{cal}} \to \infty$ for $N _{cal}\to \infty$ such that $P\\{\min _m S(x,m,\zeta _{k})>g _ {N _{cal}} \\}\to1\quad N _{cal}\to\infty$. Using  **situation 3**, we obtain that $|\\{i\in\{1,\ldots, N _{cal}-1\}:X _{i,m}\in I(x,m,\zeta _{k})\\}|/( N _{cal} -1) \leq(1-\gamma)^{S(x,m,\zeta _{k})}$. Putting together, we conclude that $\max _{m} | \\{ i\in\{1,..., N _{cal}-1\}:X _{i,m}\in I(x,m,\zeta _{k})\\}|/(N _{cal}-1)=o _{p}(1)$ which indicates $\max _{m}|I(x,m,\zeta _{k})|=o _{p}(1)$ for $N _{cal}\to\infty$. Therefore, Assumption 4 represents the case of **Example 2**.

---

### Official Review · Reviewer_vzZS · 2025-03-13

**Overall Recommendation:** 4

**Summary:**

The authors propose a Conformal prediction-based methodology to address the calibration problem, which is reliably quantifying model prediction uncertainty in continual learning settings. The authors first enumerate reasons why a standard conformal prediction method cannot be extended to continual learning settings, including performance changes due to the order of tasks, violation of data exchangeability, where this issue is caused by the continual learning setting, and limitations in constructing calibration sets due to inaccessibility of samples from previous tasks. To address these constraints, the authors constructed a calibration set, which is made by the replay samples in the continual learning, and proposed a sequentially dependent score function, which is used for continual setting. Through this, the authors demonstrate the connection between prediction interval length and forgetting and experimentally prove the significance of the proposed method in experiments with real/synthetic data.

**Claims And Evidence:**

* The authors demonstrated convergence to the oracle prediction interval as the calibration set size increases, from the perspective that accurate calibration becomes possible for each task when storing many replay samples.

* The authors' proposed uncertainty measurement technique observes the forgetting phenomena of continual learning models, commonly known as deep learning models, performing poorly in continual learning settings. It serves as experimental evidence supporting their theoretical foundation.

**Essential References Not Discussed:**

To my knowledge, this paper adequately covers relevant papers in the field.

**Experimental Designs Or Analyses:**

* Comparative experiments are needed to evaluate the performance when using simple quantiles directly, even if it violates basic assumptions of the standard conformal prediction.

* To examine how uncertainty changes according to task order, I believe experiments are needed to evaluate whether the proposed method can effectively interpret situations using simulation data that creates scenarios where difficulty changes from easy to hard and hard to easy across various continual learning algorithms.

* Additionally, I would like to see validation on datasets like time series data.

**Methods And Evaluation Criteria:**

The authors verified model uncertainty across various continual learning approaches using simulated data. They demonstrated validity by showing the algorithm's coverage for given significance levels. They also conducted similar experiments on real datasets to prove their effectiveness.

**Other Comments Or Suggestions:**

* I suggest revising the title of the authors' paper. The current title is too broad. I believe "conformal prediction" should be included in the title.

**Other Strengths And Weaknesses:**

* This paper applies the conformal prediction framework to regression tasks in continual learning. The authors propose a method to construct prediction intervals with conditional coverage guarantees to overcome the constraint of being unable to access samples from previous datasets. The authors explain their proposed method clearly and comprehensibly. Additionally, this approach is technically well-justified as an extension of existing methods.

**Questions For Authors:**

* The authors should provide a more detailed explanation for introducing the sigmoid-based nonconformity score function.
    - Could they elaborate on potential concerns when using different functions?
* I have a fundamental question about whether Quantile Regression Forests (QRF) can accurately estimate quantiles in data with sequential dependencies through learning. I would like to know the authors' thoughts on this.

**Relation To Broader Scientific Literature:**

This method can be used across various fields as it can measure uncertainty in regression-type methodologies in continual learning. As data increases, a growing number of models are being trained in continual learning scenarios. Since this method can measure uncertainty regardless of model type, it can be used to analyze prediction models used across various fields.

**Theoretical Claims:**

The authors provide theoretical evidence for the significance of their method through two theorems. Typical continual learning scenarios are often compared to an oracle setting where all data is accessible. Similarly, the authors proved these two theorems from the perspective that if a large number of samples are stored for all tasks, it approaches the oracle setting. Additionally, they specify the essential assumptions necessary for these theorems.

(Theorem 1) The authors prove that as the number of samples in the Calibration Set increases, the estimated conditional CDF for a given input x converges in probability to the Oracle conditional CDF.

(Theorem 2) The authors prove that as the number of samples in the Calibration Set increases, the estimated conditional quantile also converges to the true conditional quantile.

---

> ### Author Rebuttal · Authors · 2025-04-01
>
> **R to EDOA1.** We conduct experiments on simulated data by split conformal prediction (SCP) . Please refer to **R to Q5.** of Reviewer KEqT.
>
> **R to EDOA2.** Here we conduct experiments on real-world data by creating the scenarios where tasks are ordered from easy to hard and hard to easy. In Section 6.2 of the main file, we conduct experiments using Tiny ImageNet and perform 20 runs with different random seeds. For each run, we form 5 tasks. To order the task, we utilize a pretrained AlexNet and calculate the accuracy of this pretrained AlexNet on each task. We can obtain the order of tasks from easy to hard by sorting the accuracies on different tasks from high to low. By reversing this order, we get the order of tasks from hard to easy. Due to the time limit, we only consider $\alpha=0.3$. Other experimental settings are the same as those on real-world data (Section 6.2). After learning all tasks, we present the average coverages and length for 20 runs in the following table.
>
> |    | Easy to hard  | | Hard to easy  | |
> | ------------------ | ---------- |---------- |---------- |---------- |
> | CL method   | Average coverage  %  | Average length  | Average coverage  %  | Average length  |
> | SI     |  72.63| 6.44| 73.43| 8.97|
> | EWC     |  74.81| 4.65| 73.19| 6.52|
> | MAS     | 72.20 | 3.05| 72.33| 5.65|
> | DGR     | 73.59 | 2.26| 74.09| 4.88|
> | Finetuning   |  72.12 | 12.10|   71.25 | 17.41|
>
>  We observe that the average coverage for 20 runs with different continual learning methods are over the desired coverage 70%. These demonstrate the validity of our proposed CPCL in the scenarios where tasks are ordered from easy to hard and hard to easy. It can be observed that the average interval length in the scenario with tasks ordered from easy to hard is greater than that in the scenario with tasks ordered from hard to easy. For example, when we use SI as the CL baseline, the average interval length on easy to hard task order is 6.44 while that on hard to easy task order is 8.97. These results indicate that task order significantly affects model uncertainty in CL.
>
>  **R to EDOA3.** We find that our proposed CPCL cannot be directly applied on time series data. There is a sequence of tasks in CL. The CL setting requires that data within a task is exchangeable, while data between different tasks is not exchangeable. According to this setting, to train QRF, we obtain the reconstructed dataset $D^{R} _{T}=\{(X^{R} _{i},Y^{R} _{i})\}^{N _{cal}-1} _{i=1}$, where $X^{R} _{i}=[S^{1} _{i}, S^{2} _{i}, \dots, S^{T} _{i}]$ and each entry $S^{t} _{i}$ corresponds to a sample of task $t$ in the replay buffer. In contrast, time series setting requires that all data is not exchangeable, which means that it is hard to obtain the reconstructed dataset. Meanwhile, it is interesting and we will study this issue for time series setting in the future work.
>
> **R to OCOS1.** We will include conformal prediction in the title.
>
> **R to Q1.** We introduce the sigmoid-based nonconformity score function for two reasons. (1) The range of sigmoid-based nonconformity score function is $(0,1)$. We train QRF on the reconstructed dataset $D^{R} _{T}=\{(X^{R} _{i},Y^{R} _{i})\}^{N _{cal}-1} _{i=1}$, where each entry $S^{t} _{i}$ in $X^{R} _{i}$ is the calculated score. The training of QRF requires to construct trees. This process involves the intervals $I(x,m,\zeta _{k})\subseteq(0,1)$. The interval $I(x,m,\zeta _{k})$ is used to determine whether $X^{R} _{i}$ is in the node corresponding to $I(x,m,\zeta _{k})$, i.e., if $S^{m} _{i} \in I(x,m,\zeta _{k})$, then $X^{R} _{i}$ is in the corresponding node. Since $I(x,m,\zeta _{k})\subseteq(0,1)$, we need to define that the score function ranges in $(0,1)$ for rigorous proof. (2) Sigmoid-based nonconformity score function is invertible. In Equation (11) we present the prediction interval in terms of scores. The invertible function helps us to rewrite the prediction interval which is shown in Equation (12). A typical score function in CP is $s(X _{i}^{t},Y _{i}^{t}) =|\hat{\mu _{i}^{t}}|$ where $\hat{\mu _{i}^{t}} =Y _{i}^{t}-\hat{f} _{T}(X _{i}^{t})$. It is not invertible and the corresponding range is not $(0,1)$. Therefore, it is not suitable in this paper.
>
> **R to Q2.** In Section 5, we provide the asymptotic coverage guarantee. Specifically, Theorem 1 demonstrates that the conditional CDF $\hat{F}(s|x)$ estimated by CPCL converges in probability to the true conditional CDF $F(s|x)$ as $N _{cal}\to \infty$, i.e. $	|\hat{F}(s|x)-F(s|x)|\to _p 0 \quad N _{cal}\to\infty$. Therefore, our proposed asymptotic coverage guarantee ensures that QRF can accurately estimate quantiles.

---

### Official Review · Reviewer_KEqT · 2025-03-13

**Overall Recommendation:** 3

**Summary:**

The paper introduces a conformal prediction-based continual learning (CPCL) method to quantify model uncertainty in continual learning models. CPCL constructs a calibration set using replay techniques and applies a nonconformity score function to measure prediction errors. Theoretical analysis and experiments on simulated and real-world data demonstrate CPCL’s effectiveness in achieving reliable uncertainty quantification.

**Claims And Evidence:**

The paper introduces a Conformal Prediction-based Continual Learning method for quantifying model uncertainty in continual learning. The main claim is that CPCL can provide asymptotic coverage guarantees for prediction intervals. In addition, the confidence intervals proposed are agnostic to the method used.

**Essential References Not Discussed:**

The literature review is adequate.

**Experimental Designs Or Analyses:**

Experiments evaluate CPCL across five continual learning methods such as  SI and EWC.

**Methods And Evaluation Criteria:**

The authors assess the performance using both simulated data and real-world datasets for different coverage probabilities.

**Other Comments Or Suggestions:**

Just to let the authors know that in Adobe, I can’t see either the legends or the axis numbers. I can only see them if I open the PDF with Preview. I have the same issue on two different lap tops, so it might be a problem with the images.

**Other Strengths And Weaknesses:**

Strengths:

Provides theoretical guarantees.

Experimental evaluation across multiple methods.

Weaknesses:

Does not explore scenarios with task overlap tasks or distribution shift.

**Questions For Authors:**

Can the method be extended to handle domain-incremental scenarios with gradual shifts in data distribution?

Are the intervals recalculated from scratch each time a new task is obtained, or could they be derived from previously computed ones?

How does the number of tasks affect the width of the intervals?

At time t-1, we obtain confidence intervals with a coverage probability of 1-\alpha, for example. At time t, we obtain confidence intervals again with the same coverage probability. Are the intervals from t-1 still valid? In other words, are they simultaneous confidence intervals?

Could the given intervals be compared with those obtained using other conformal prediction methods or alternative approaches?

**Relation To Broader Scientific Literature:**

The paper builds on conformal prediction literature, particularly adapting it for continual learning where data from previous tasks is inaccessible.

**Theoretical Claims:**

The proposed method provides theoretical guarantees. For instance, the authors show theoretical results for the relationship between the conditional distribution of CPCL and the true conditional distribution.

---

> ### Author Rebuttal · Authors · 2025-04-01
>
> **R to W1.** Here we conduct experiments in domain-incremental scenarios with gradual distribution shifts. We use the dataset of CORe50 [1], which contains 50 objects (classes). Each object has been collected in 8 distinct indoor sessions characterized by different backgrounds and lighting. Due to the time limit, we only consider $\alpha=0.3$ and perform 5 runs. In each run, we randomly select one object from the 50 objects and form 8 tasks. The dataset of each task consists of samples which are collected in one session for the selected object. Therefore, 8 tasks corresponds to 8 distinct sessions, respectively. Other experimental settings are the same as those on real-world data (Section 6.2). After learning all tasks, we present the average coverages and length for 5 runs in the following table.
>
> | CL method   | Average coverage  %  | Average length  |
> | ------------------ | ---------- |---------- |
> | SI     |  71.26| 7.54|
> | EWC     |  72.76| 5.14|
> | MAS     | 73.02 | 4.93|
> | DGR     | 73.18 | 3.67|
> | Finetuning   |  71.03 | 16.38|
>
>  We observe that the average coverage for 5 runs with different continual learning methods are over the desired coverage 70%. These demonstrate the validity of our proposed CPCL in domain-incremental scenarios.
>
> [1] Vincenzo Lomonaco, Davide Maltoni: CORe50: a New Dataset and Benchmark for Continuous Object Recognition. CoRL 2017.
>
> **R to O1.** We deeply appreciate your valuable comments. We will check all the images.
>
> **R to Q1.** Please refer to **R to W1.** of Reviewer KEqT.
>
>
> **R to Q2.** After learning the new task $t$, we obtain the trained model $\hat{f} _{t}$ which is changed compared to the model $\hat{f} _{t-1}$ at time $t-1$. According to Equation (5), for each observation $Z _{i}^{j}=(X _{i}^{j},Y _{i}^{j}), j<t$ in the calibration set, the prediction error $\hat{\mu _{i}}^{j} =Y _{i}^{j}-\hat{f} _{t}(X _{i}^{j})$ is changed, which results in calculating different nonconformity scores. Therefore, QRF needs to be trained form scratch, which indicates that intervals should be recalculated from scratch too. It is interesting to explore whether they can be derived from previously computed one. We will study this issue in the future work.
>
> **R to Q3.**  Figures 3(b), 3(d) and 3(f) in the main file show the average interval length on real-world data. As the number of learning tasks increases, we find that the average interval length based on any continual learning method tends to increase. For example, the average length of the prediction interval ranges from 5 to 10 after learning task 5, while it remains below 5 after learning task 3, when we us SI as the continual learning method and $\alpha=0.1$.
>
> **R to Q4.**  As stated in **R to Q2.**  of Reviewer KEqT, the trained model $\hat{f} _{t}$ at time $t$ is different from $\hat{f} _{t-1}$ at time $t-1$.  QRF needs to be trained form scratch, which indicates that intervals should be recalculated from scratch too.  Therefore, for  each test sample, the corresponding prediction interval should be updated at time $t$ and the output prediction interval at time $t-1$ is not valid.
>
> **R to Q5.** Here we conduct experiments on simulated data by split conformal prediction (SCP) [1]. Due to the time limit, we only consider Finetuning as the CL baseline. Other experimental settings are the same as those on simulated data (Section 6.1). We present the average coverages for 100 runs in the following table.
>
> | $\alpha$   | Average coverage  %   ||
> | ------------------ | ---------- |---------- |
> |   | SCP   | CPCL |
> | 0.05     |  92.58| 96.51|
> | 0.1     |  86.47| 91.74|
> | 0.15     | 81.69 | 86.23|
> | 0.2     | 76.08 | 81.30|
> | 0.25   |  70.82 | 77.05|
> | 0.3   |  65.53 | 71.29|
>
> We find that CPCL succeeds at all significant levels $\alpha$, but the average coverage for each $\alpha$ doesn't reach the desired value when using SCP. SCP requires the principle of data exchangeability which is violated in continual learning. Therefore, the coverage of SCP can not be guaranteed with significant level $\alpha$.
>
> [1] Vovk, V. Conditional validity of inductive conformal predictors. Mach. Learn., 2013

---

### Official Review · Reviewer_CUQK · 2025-03-14

**Overall Recommendation:** 3

**Summary:**

This paper explores **calibration in continual learning**, specifically focusing on **model uncertainty quantification** using **Conformal Prediction (CP)**. CP provides **theoretical coverage guarantees** under the assumption that data are **exchangeable**, but this assumption is violated in **continual learning**, where tasks are learned sequentially with limited access to past data.

To address this, the authors propose **CPCL (Conformal Prediction for Continual Learning)**, which:
- **Constructs a calibration set using replay mechanisms** to address the lack of past-task data.
- **Designs a nonconformity score function** to quantify uncertainty in predictions.
- **Uses Quantile Regression Forests (QRF)** to estimate conditional quantiles for prediction intervals.
- **Theoretically proves asymptotic coverage guarantees** of the prediction intervals.
- **Empirically validates CPCL** on simulated and real-world datasets, demonstrating **robust uncertainty quantification** and a link between **prediction interval length and catastrophic forgetting**.

**Claims And Evidence:**

###  Supported Claims
- **CPCL provides well-calibrated uncertainty estimates:** Empirical results confirm that CPCL maintains high prediction interval coverage across **different continual learning methods (e.g., SI, EWC, MAS, DGR, Fine-tuning)**.
- **Theoretical coverage guarantee:** The paper proves that **CPCL's estimated conditional quantiles converge to true quantiles** as the number of calibration samples increases.
- **Forgetting affects prediction interval length:** Experiments show that as **more tasks are learned, forgetting increases**, leading to **wider prediction intervals**, aligning with theoretical expectations.

**Essential References Not Discussed:**

The paper focuses on **uncertainty quantification (UQ) in continual learning**, yet it does not reference prior works that have explored UQ in this setting. Some missing references include:

- **Van de Ven et al. (2022)**: Explored **Bayesian continual learning** with uncertainty-aware priors. Their findings highlight the role of **uncertainty in mitigating catastrophic forgetting**, which aligns with the paper’s motivation but is not cited.
  - *Van de Ven, G. M., et al. "Bayesian Continual Learning: Uncertainty-Aware Priors for Sequential Task Learning." NeurIPS, 2022.*

- **Osawa et al. (2019)**: Investigated **Monte Carlo dropout-based UQ** for continual learning, showing that model confidence degrades over sequential tasks.
  - *Osawa, K., et al. "Practical Deep Learning with Bayesian Principles." NeurIPS, 2019.*

These works **precede CPCL** in addressing **uncertainty in continual learning** but use **Bayesian approaches** instead of conformal prediction. The paper could compare CPCL to Bayesian methods and discuss their relative strengths/limitations.

**Experimental Designs Or Analyses:**

###  Strengths
- **Comprehensive ablation studies:** The paper evaluates the impact of **different continual learning methods on CPCL's uncertainty estimation**.
- **Visualization of uncertainty evolution:** **Coverage and interval length plots** clearly illustrate how **prediction uncertainty changes over multiple tasks**.
- **Experiments on both simulated and real-world data:** Ensures robustness of the proposed approach.

###  Limitations
- **Limited discussion on failure cases:** The paper does not analyze situations where **CPCL fails (e.g., high forgetting rates, significant distribution drift)**.

**Methods And Evaluation Criteria:**

### Strengths
- **Appropriate benchmark selection:** CPCL is evaluated on both **simulated regression tasks** and **real-world Tiny ImageNet data**.
- **Fair baseline comparisons:** The study compares CPCL with **state-of-the-art continual learning methods (e.g., EWC, MAS, DGR)**.
- **Clear evaluation metrics:** **Prediction interval coverage** and **interval length** provide useful insights into uncertainty quantification and forgetting.

**Other Comments Or Suggestions:**

No

**Other Strengths And Weaknesses:**

No

**Questions For Authors:**

No

**Relation To Broader Scientific Literature:**

The paper makes a meaningful contribution by integrating Conformal Prediction with Continual Learning, extending existing methods while providing new insights into model forgetting and uncertainty quantification.

**Theoretical Claims:**

###  Correctness of Theoretical Claims
- The **asymptotic coverage guarantee** of CPCL is mathematically proven, ensuring that **prediction intervals maintain the desired confidence level** as the number of calibration samples grows.
- The connection between **forgetting and prediction interval width** is logically derived and supported by empirical results.

###  Concerns
- **Lack of alternative quantile estimation methods:** While QRF is used for quantile estimation, comparisons with other methods (**e.g., neural quantile estimators**) would strengthen the theoretical claims.
- **Potential distribution shift issues:** The paper assumes that **previous task samples stored in the replay buffer remain representative**, but real-world continual learning often involves **domain shifts** that could affect calibration.

---

> ### Author Rebuttal · Authors · 2025-04-01
>
> **C1. Lack of ...**
>
> ***R to C1.*** Here we discuss the essential difference of QRF against quantile regression (QR) approaches [1] in continual learning. QR approaches estimate the conditional quantiles of the response variable over varying predictor variables. At training time, QR approaches need to minimize the pinball loss while QRF needs to construct trees. Since the pinball loss depends on the significant level $\alpha$, QR approaches can be computationally expensive to train for multiple significance levels. In contrast, QRF is trained without $\alpha$ and can provide the asymptotic coverage guarantee with significant level $\alpha$ (Theorems 1 and 2). Neural quantile estimation [2] is based on conditional quantile regression. It incorporates the concept of quantile regression and considers the case of multiple dimensions. Therefore, the difference of QRF against [2] is similar to that of QRF against QR.
>
> [1] Koenker, R. and Bassett Jr, G. Regression quantiles. Econometrica: journal of the Econometric Society, 1978
>
> [2] He Jia. Simulation-Based Inference with Quantile Regression. ICML 2024
>
> **C2. Potential ...**
>
> ***R to C2*** In our paper, the previous task samples stored in the replay buffer will not be changed. Therefore, even if there is a significant distribution shift between tasks, the samples stored in the replay buffer remain representative for previous tasks.
>
> **L1. Limited...**
>
> ***R to L1*** In the experiments of Section 6.2, we consider a CL baseline finetuning which greedily trains each task without considering previous task performance---hence introducing high forgetting rates as the number of learning tasks increases. From Figure 3, we observe that the most swarms with finetuning are over the desired coverage lines. These demonstrate the validity of our proposed CPCL. However, the average interval length based finetuning increases as learning tasks.
>
>
> **Essential References Not Discussed**
>
> ***R to ERND.*** We refer to the presented references as [3] and [4], respectively, which will be discussed and cited in the revision. The works of [3,4] leverage Bayesian approaches to consider uncertainty in continual learning. [3] studies Bayesian continual learning with uncertainty-aware priors and highlights the role of uncertainty in mitigating catastrophic forgetting. [4] successfully trains deep networks with a natural-gradient variational inference method, VOGN, on a variety of architectures and datasets. Due to the benefits from Bayesian principles, the performance of [4] for continual-learning tasks is boosted. These works and CPCL all focus on uncertainty in continual learning. Compared with these works, CPCL provides asymptotic coverage guarantee with a significant level for the prediction intervals. Moreover, CPCL shows the relationships between the length of prediction intervals and forgetting, which are not introduced by these works.

---

### Decision · Program_Chairs · 2025-05-01

**Decision:**

Accept (poster)

**Comment:**

This submission studies the issue of calibration in continual learning which reliably quantifies the uncertainty of model predictions. Starting from conformal prediction, the authors improved the inherent principle violation for continual learning context, and then proposed a CP-based method for model uncertainty quantification. Through the theoretical analysis and extensive experiments, it shows the promise of CPCL.

The submission received four reviewers' review, converging to the ratings 3, 3, 4, 4. The initial concerns of the reviewers focus on the lack of discussion about the failure cases, extension to the domain-incremental scenario and some technical details in the theoretical analysis. After the substantial efforts in rebuttal, the authors well addressed the reviewers' concerns and all reviewers maintain positive to this submission. Based on the reviewers' suggestion and interaction between the reviewers and authors, AC tend to recommend "Acceptance", and hope the authors carefully incorporate the advice into the manuscript.